# Cognitive Assessment Based on Electroencephalography Analysis in Virtual and Augmented Reality Environments, Using Head Mounted Displays: A Systematic Review

**Foteini Gramouseni** [1], **Katerina D. Tzimourta** [1], **Pantelis Angelidis** [1], **Nikolaos Giannakeas** [2] **and Markos G. Tsipouras** [1,*]

1 Department of Electrical and Computer Engineering, University of Western Macedonia, 501 00 Kozani, Greece; f.gramouseni@uowm.gr (F.G.); ktzimourta@uowm.gr (K.D.T.); paggelidis@uowm.gr (P.A.)
2 Department of Informatics and Telecommunication, University of Ioannina, 451 10 Ioannina, Greece; giannakeas@uoi.gr
* Correspondence: mtsipouras@uowm.gr

**Abstract:** The objective of this systematic review centers on cognitive assessment based on electroencephalography (EEG) analysis in Virtual Reality (VR), Augmented Reality (AR) and Mixed Reality (MR) environments, projected on Head Mounted Displays (HMD), in healthy individuals. A range of electronic databases were searched (Scopus, ScienceDirect, IEEE Explore and PubMed), using PRISMA research method and 82 experimental studies were included in the final report. Specific aspects of cognitive function were evaluated, including cognitive load, immersion, spatial awareness, interaction with the digital environment and attention. These were analyzed based on various aspects of the analysis, including the number of participants, stimuli, frequency bands range, data preprocessing and data analysis. Based on the analysis conducted, significant findings have emerged both in terms of the experimental structure related to cognitive neuroscience and the key parameters considered in the research. Also, numerous significant avenues and domains requiring more extensive exploration have been identified within neuroscience and cognition research in digital environments. These encompass factors such as the experimental setup, including issues like narrow participant populations and the feasibility of using EEG equipment with a limited number of sensors to overcome the challenges posed by the time-consuming placement of a multi-electrode EEG cap. There is a clear need for more in-depth exploration in signal analysis, especially concerning the $\alpha$, $\beta$, and $\gamma$ sub-bands and their role in providing more precise insights for evaluating cognitive states. Finally, further research into augmented and mixed reality environments will enable the extraction of more accurate conclusions regarding their utility in cognitive neuroscience.

**Keywords:** EEG; cognition; virtual reality; augmented reality; mixed reality; systematic review; PRISMA

## 1. Introduction

Cognition is one of the most fundamental research topics of the neuroscience field. The last decade has seen an increase in interest in evaluating the impact of a system on users' cognitive state. A cognitive function is a broad term that refers to "mental processes involved in the acquisition of knowledge, manipulation of information and reasoning" [1]. The evaluation of cognitive status is typically carried out through specifically designed questionnaires. Nevertheless, this assessment approach encompasses the subjective element, which can potentially lead to misleading conclusions.

Evidence has been presented that electroencephalogram (EEG) oscillations reflect cognitive and memory performance, and they can be used as a viable method for cognitive assessment. Electroencephalography is a non-invasive method to record the electrical activity of the brain, through a number of electrodes placed on the scalp. It has been shown that

electrical activity represents neural cognitive and emotional functions. Intracranial EEG, also called Electrocorticography (ECoG), involves invasive electrode implants, surgically placed directly on the surface of the brain. This review does not include research using invasive methods for EEG signal acquisition. The bandwidth of the EEG signals ranges between 0.5–40 Hz. The five most commonly studied rhythms are the following:

- delta ($\delta$) (0.5–4 Hz)—associated with deep sleep
- theta ($\theta$) (4–8 Hz)—observed during quiet focus and sleep
- alpha ($\alpha$) (8–14 Hz)—recorded with closed eyes and relaxing
- beta ($\beta$) (14–30 Hz)—associated with alertness and attentional allocation
- gamma ($\gamma$) (over 30 Hz)—linked with learning and high mental state

However, a significant number of researchers further subdivide the above-mentioned frequency bands into sub-bands. According to this approach, $\alpha$, $\beta$, and $\gamma$ bands can be further divided into:

- $\alpha1$ (7–10 Hz)—associated with a relaxed but alert state
- $\alpha2$ (10–13 Hz)—linked to more active cognitive processing than $\alpha1$
- $\beta1$ (13–18 Hz)—associated with active, attentive cognitive processing
- $\beta2$ (18–30 Hz)—associated with more complex cognitive processes
- $\gamma1$ (30–40 Hz)—linked to sensory processing and perception
- $\gamma2$ (40–50 Hz)—involved in higher-level cognitive processes and feature binding
- $\gamma3$ (50–80 Hz)—useful for research focused on exploring the synchronization of neural networks and its role in various cognitive functions

When designing their experiments, researchers often encounter difficulties in holding the trials either in real life or in a laboratory environment. Digital environments offer an excellent solution to this problem, promising to fill the gap between the physical and the artificial environment. The Encyclopedia Britannica [2] describes VR as "the use of computer modeling and simulation that enables a person to interact with an artificial three-dimensional (3D) visual or other sensory environment". According to the same source, AR [3] is defined as "a process of combining or "augmenting" video or photographic displays by overlaying the images with computer-generated data", while MR is a new technology, that combines both VR and AR experiences.

A distinction between low-immersion and high-immersion systems is typically made in the literature. In low-immersion systems, the digital environment is displayed on a conventional 2D screen and the interaction is controlled through a computer mouse or a keyboard. In the case of AR low-immersion systems, often called see-through environments, a mobile device is used to watch 3D visualizations on the scene, recorded by the device camera. In highly immersive systems, a head-mounted-display (HMD) is often used. HMDs are high-resolution display devices, fixed on one's head and mounted in front of the eyes, often combined with headsets or haptic feedback. The interaction is controlled through motion sensors connected with a computer system, so the field of view can be changed according to the users' head movement, while exploring the digital environment [4,5]. However, a very serious drawback in conducting experiments through VR is the sensation of dizziness and nausea, known as VR sickness, often experienced by participants. This phenomenon can significantly affect cognitive state and, consequently, the experiment's results. This is one of the key reasons why research involving VR is not wide spread and has a restricted number of participants. This review is focused on studies with stimulus presented on HMD devices.

Recent research has focused on specific cognitive functions that can be categorized into the following fields: cognitive load, immersion, spatial awareness, interaction with the Digital Environment and attention. Interaction with the digital environment is a general category with research in the fields of cognitive conflict, performance, communication, affordance and creativity as shown in Figure 1.

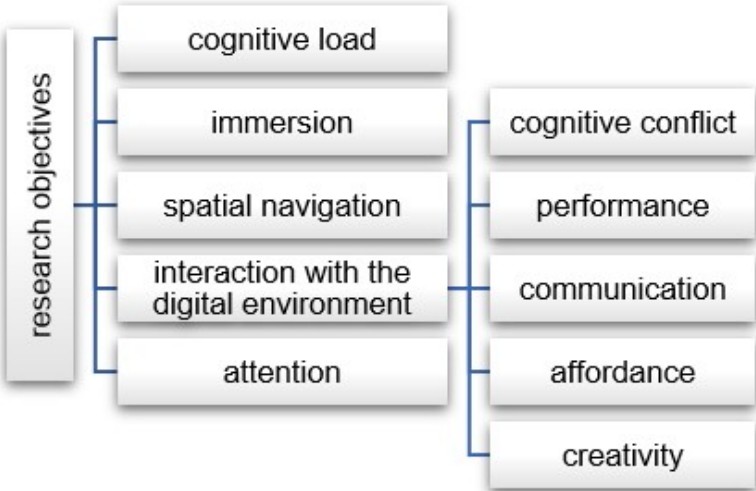

**Figure 1.** Research objectives.

Cognition is a field that has been investigated by many neuroscientists, but to date, there is no systematic review that reviewed cognitive assessment based on EEG analysis in digital environments using HMDs. A synthesis of recent research in this domain will foster a more complete understanding of the nature of cognition in VR, AR and MR environments, and serve as a guide to future research in the field. In the sections below, research stimulus and outcomes are analyzed. The methods for data preprocessing, artifact removal, frequency bands and statistical analysis or classification methods on EEG data recordings are also investigated.

## 2. Research Methodology

The methodological approach used to obtain relevant studies follows the Preferred Reporting Items for Systematic Review and Meta-Analysis (PRISMA) [6]. More than two independent researchers worked through the screening process of the articles for eligibility and they also assessed the risk of bias of all included studies. All disagreements were resolved by discussion consensus between the reviewers. The review protocol has been registered with the Open Science Framework [7] (url: https://osf.io/kfx5p, accessed on 2 March 2023).

### 2.1. Data Sources

Electronic search was performed on December 2022 using the following databases: Scopus, ScienceDirect, IEEE Explore and PubMed. The following keyword combinations were employed in the search: (("EEG" OR "electroencephalography") AND ("augmented reality" OR "virtual reality" OR "mixed reality") AND ("cognition" OR "cognitive")), that have been published during the last ten years, from 2013 to 2022. Rayyan [8],a free web-tool designed to help researchers working on reviews, was used for the screening of the total 2326 results extracted from the database search. After duplicate removal, title and abstract screening, 427 articles remained for full text review. A detailed analysis of the screening process is listed in Figure 2.

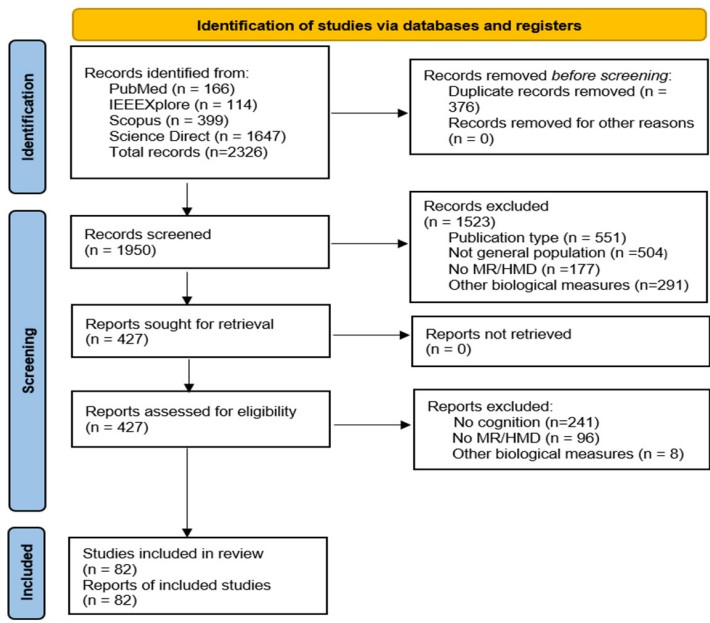

**Figure 2.** PRISMA flow chart.

*2.2. Inclusion Criteria*

This systematic review includes articles providing results from:

- randomized controlled trials original primary research
- healthy participants from general population without pathological history or any kind of disorders
- a Head Mounted Display device (HMD) as the stimuli projection system
- EEG signals as the only neuroimaging measure
- at least one EEG-assessed cognition related research topic (i.e., cognitive load, immersion, spatial awareness, interaction with the digital environment, attention)
- at least one EEG-based neurobiological outcome

*2.3. Exclusion Criteria*

A number of articles have been excluded from this review, under the following criteria:

- conference articles and case studies
- theoretical studies, such as review articles, overviews, meta-analyses and book chapters
- research conducted on animals
- published in language other than English
- including participants with pathological history (e.g., Alzheimer's disease, Parkinson, post-stroke patients, brain injury, autism, epilepsy, visual or cognitive impairment/decline, disabilities, etc.);
- including participants with disorders (e.g., alcoholism, attention disorder, anxiety disorder, psychosis, pathological gambling, etc.);
- including participants from expert groups (e.g., skiers, pilots)
- using biological measures other than EEG for the research outcomes (e.g., functional Magnetic Resonance Imaging (fMRI), Electrocardiogram (ECG), Electrooculogram (EOG), Electromyogram (EMG), Galvanic Skin Response (GSR), Heart Rate (HR), Electrocardiogram (EKG))
- with research objectives not in the field of cognition (e.g., emotion, pain, sleep, motor functions)
- not including VR, AR or MR stimuli
- with stimulus displayed on devices other than HMD (e.g., projectors, screens or specially designed spaces)
- not reporting results (e.g., study protocols, datasets).

*2.4. Data Synthesis*

A total of 82 studies met the inclusion criteria and the following data were extracted and analyzed: publication year, total number of participants, digital environment type (AR, VR or MR), EEG equipment type and electrode number. The papers were grouped into five categories according to their objective area: cognitive load, interaction with the digital environment and attention, as shown in Figure 1. For each category, objectives, outcomes, frequency bands, preprocessing and artifact removal methods, classification methods and statistical analysis methods on EEG data are examined.

## 3. Study Statistics

*3.1. Publication Year*

Included articles were published from 2013 until December 2022, with incremental number of publications during the years. Table 1 presents the number of studies for each publication year.

**Table 1.** Publication Year.

| Publication Year | # of Studies |
|:---:|:---:|
| 2013 | 0 |
| 2014 | 1 |
| 2015 | 1 |
| 2016 | 2 |
| 2017 | 3 |
| 2018 | 9 |
| 2019 | 14 |
| 2020 | 13 |
| 2021 | 16 |
| 2022 | 23 |
| Total | 82 |

*3.2. Total Number of Participants*

The mean number of participants across all included articles was 34.8 (SD = 45.2, range = 1–340). Table 2 illustrates the distribution of participants.

**Table 2.** Number of participants distribution.

| Participants | # of Studies |
|:---:|:---:|
| 1–10 | 8 |
| 11–20 | 23 |
| 21–30 | 25 |
| 31–40 | 9 |
| 41–60 | 9 |
| >60 | 8 |
| Total | 82 |

*3.3. Digital Environment Type (VR, AR, MR) and Equipment*

A wide range of HMD types exists, including see-through styles, mobile designs, camera-attached AR styles, glass-based configurations, options with eye-tracking capabilities, and varying resolutions. Table 3 shows the type of digital environment and the equipment used to present the stimulus in the experimental tasks. The overwhelming majority of tasks utilized VR (Figure 3).

**Table 3.** Digital environment type and equipment.

| Digital Environment Type | Device | References | # of Studies |
|---|---|---|---|
| VR | Samsung Gear VR | [5] | 74 |
| | HTC Vive | [9–38] | |
| | HTC Vive Pro | [39–46] | |
| | HTC Vive Focus | [47–49] | |
| | Oculus Rift DK2 | [50–57] | |
| | Oculus Rift | [58–63] | |
| | Oculus Rift S | [64] | |
| | Oculus | [65,66] | |
| | Oculus Go | [67] | |
| | nVisor SX60 | [68–70] | |
| | 3DVR | [71] | |
| | HTC Vive/Samsung Odyssey | [72] | |
| | MIUI PLAY2 | [73] | |
| | Silicon Micro Display ST1080-10V1 | [74] | |
| | VIVE-P130 | [75] | |
| | ACER WMR | [76] | |
| | n/a | [77–81] | |
| AR | Hololens | [82,83] | 8 |
| | Hololens 2 | [84] | |
| | Sony SmartEyeglass SED-E1 | [85,86] | |
| | DreamWorld AR | [87] | |
| | Vuzix Wrap 1200DXAR | [88] | |
| | n/a | [89] | |
| Total | | | 82 |

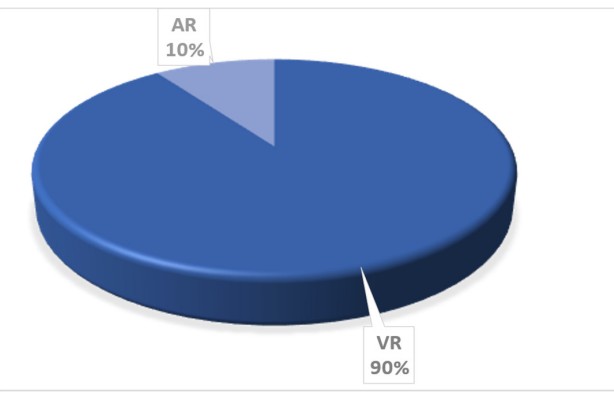

**Figure 3.** Digital environment type.

*3.4. EEG Equipment*

EEG recording systems use wet or dry electrodes that must be in contact with the scalp. A traditional multi-electrode system is usually fixed on the head using a cap or a helmet, which helps to ensure the precise placement of the electrodes and their sufficient contact with the scalp. More recent EEG technologies with limited number of electrodes have been developed, where electrodes are fixed on a headband or a headset, providing an easy-to-use setting. Table 4 displays a categorized list of EEG equipment utilized in the study. As illustrated in Figure 4, the vast majority of the studies utilized a multi-electrode EEG equipment.

**Table 4.** EEG equipment.

| EEG Type | | References | # of Studies | Percentage |
|---|---|---|---|---|
| Cap Patches Helmet | ABM X-10 | [5,15,16,20] | 71 | 87% |
| | G-Tec | [9] | | |
| | ASA Lab, ANT | [51,68] | | |
| | Nihon Kohden | [39] | | |
| | LiveAmp | [10,44] | | |
| | g.LADYBIRD | [11,12,50] | | |
| | g.GAMMAsys | [72] | | |
| | g.USBamp | [36] | | |
| | g.tec Nautilus | [82] | | |
| | Brain Products | [77] | | |
| | QuickAmp | [79] | | |
| | actiCHamp | [62,63] | | |
| | BrainAmp | [46] | | |
| | BrainAmp Move System | [54,76] | | |
| | Enobio 3 | [13] | | |
| | Enobio-32 | [88] | | |
| | StarStim 8 | [35] | | |
| | EPOC Flex | [14] | | |
| | Neuracle | [17,18] | | |
| | OpenBCI | [47–49] | | |
| | Biosemi Active Two | [21] | | |
| | Biosemi Actiview | [42,43] | | |
| | V-Amp | [22] | | |
| | ANT Neuro | [52,53,64] | | |
| | eegoSports | [81] | | |
| | ActiCAP | [25,28,57,67,70,78] | | |
| | BIOPAC MP160 | [75] | | |
| | Mobita | [83] | | |
| | EasyCap | [32,41,85,86] | | |
| | Scan SynAmps2 Express | [30] | | |
| | Curry 8 SynAmps2 Express | [31] | | |
| | Neuracle | [74,80] | | |
| | mBrainTrain | [87] | | |
| | Nuamps7181 | [84] | | |
| | B-Alert | [33,37] | | |
| | n/a | [19,24,27,29,38,60,65,69,74,80] | | |
| Headband Headset | QUASAR DSI-7 | [58] | 11 | 13% |
| | Looxid Link | [40] | | |
| | EMOTIV EPOC+ | [34,59,66] | | |
| | NeuroSky MindWave | [23,45,71] | | |
| | MUSE | [26] | | |
| | MyndPlay BrainBand XL | [56] | | |
| | n/a | [89] | | |

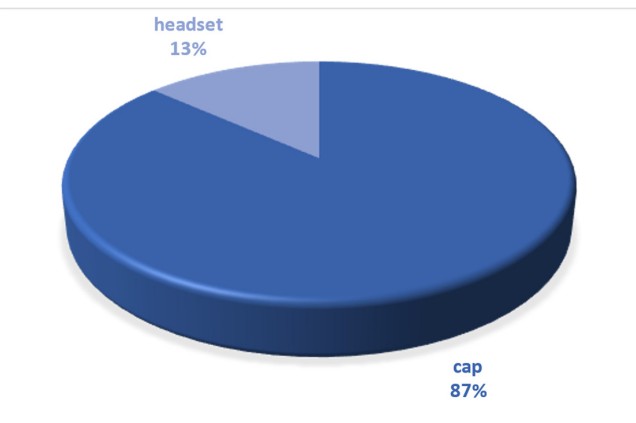

**Figure 4.** EEG type.

### 3.5. Number of Electrodes Used

Not all studies process the signals acquired from the whole electrode set from the EEG equipment used in their experiments. It usually depends on the Regions of Interest (ROIs) they have determined that play the most significant role in their objective. Table 5 displays the distribution of studies based on the number of electrodes used. Figure 5 displays the number of electrodes used, categorized into four groups. It should be noted that the number of electrodes is not reported for all articles.

**Table 5.** Distribution of number of electrodes utilized by researchers.

| # of Sensors | References | # of Studies |
|:---:|:---:|:---:|
| 1–4 | [23,26,45,56,71,79,89] | 7 |
| 6–14 | [5,10–13,15–17,20,22,35–37,40,47–49,58,59,65,81,84,85,88] | 24 |
| 16–35 | [9,14,19,24,27,30,33,34,39,41,44,46,55,57,60,66,67,72,73,75,77,82,83,86,87] | 25 |
| >57 | [18,21,25,28,29,31,32,38,42,43,50–54,61–64,68–70,74,76,78,80] | 26 |

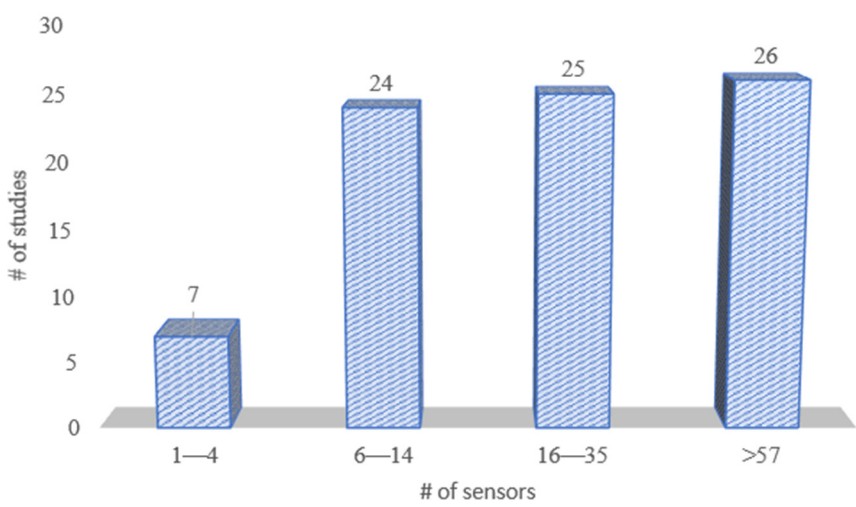

**Figure 5.** Number of sensors distribution.

### 3.6. Objective Area

The majority of researchers implemented experiments focusing on cognitive load analysis, many studies investigated the participants' interaction with the digital environments and attention, while a few papers assessed the immersion state and spatial awareness of the subjects (Table 6).

**Table 6.** Objective area.

| Objective | References | # of Studies |
|---|---|---|
| Cognitive load | [5,9–20,39–41,47–49,51,58,76,77,84–86,89] | 27 |
| Immersion | [14,21–23,50,59,60,65,74,88] | 10 |
| Spatial awareness | [24,25,42,43,52–54,61,68,71,78] | 11 |
| Interaction with the digital environment | [23,26,28–32,44,55,62,64,67,69,72,79–81,83] | 19 |
| Attention | [33–38,45,46,56,57,63,66,70,73,75,82,87] | 17 |

It should be noted that a number of studies included objectives from more than one field, but in many cases their results came from subjective analysis (i.e., questionnaires) or from measurements other than EEG signals, for example, the time spent in a virtual room or the number of successful targets hit, etc. In this review, only the results that come from EEG analysis are reported. Figure 6 provides a visual representation of the distribution of studies across different objectives.

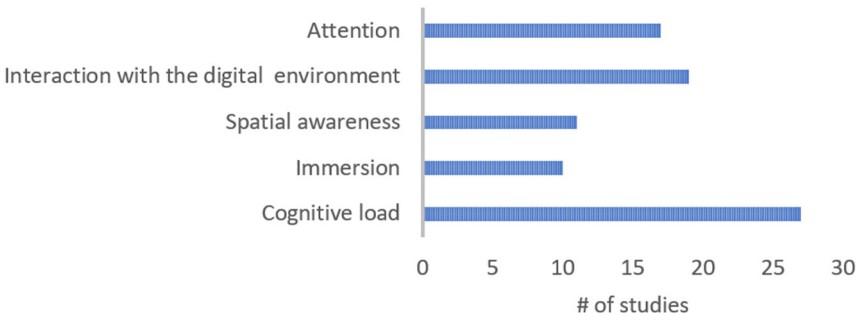

**Figure 6.** Objective area.

## 4. Results

### 4.1. Paper Layout

For each section a comparative analysis is presented for the objectives and outcomes, the data preprocessing and artifact removal methods, the frequency bands and the classification or statistical analysis on EEG recordings the authors have implemented in their research. Both statistically significant and non-significant results are included to avoid risk of bias in the synthesis of the results.

### 4.2. Cognitive Load

Cognitive load, also known as information flow, refers to the quantity of data that can be retained by working memory simultaneously. According to [90], cognitive load theory can provide guidelines to support the delivery of information in a way that stimulates learner engagement, leading to the maximization of cognitive performance. Table 7 presents a summary of findings in the field of cognitive load research.

**Table 7.** Cognitive load.

| Authors, Year, Reference | Participants | Stimuli | Frequency Bands/ Range | Data Preprocessing | Artifact Removal | Classification Technique/Statistical Analysis | Main findings |
|---|---|---|---|---|---|---|---|
| (Seeling, 2017) [89] | 30 | VIEW dataset | θ, α, β | | | average α/θ, α/β average and variability, α and θ variability, R2, KNN | • the α/β ratio provides a good average estimation of the cognitive load levels<br>• variable accuracies for individual cognitive load assessment |
| (Gerry et al., 2018) [9] | 2 | visual search task | α | notch, low pass | | central tendency of α, ERD | • increased complexity was correlated with decreased α power |
| (Ikiz et al., 2019) [85] | 4 | automobile assembly line | β, γ | notch | BrainVision Analyzer 2 software | EEG graph area, One Sample *t*-test | • no extra cognitive burden was associated with utilizing AR glasses<br>• no age-related effects |
| (Makransky et al., 2019) [5] | 52 | text PC vs. VR, with/no narration | 1–40 Hz | | ABM's proprietary software | LDFA, stepwise regression, ANOVA | • participants learned less in the VE<br>• significantly higher cognitive load in the VE<br>• no significant differences when adding narration |
| (Kakkos et al., 2019) [51] | 29 | flight simulator (2D vs. VR) | δ, θ, α, β, γ | 0.5–40 Hz, notch | ICA | eLORETA, AAL-116, PLI, EG, EL, RFE-CBR, 10-fold cross-validation, LDA, ANOVA | • elevated mental workload in 3D interfaces<br>• higher connectivity strength in VR condition |
| (Qadir et al., 2019) [39] | 11 | driving simulator (2D vs. VR) | θ, α1, α2, β | | CAR, EllipticalBPF, ICA, EEGLAB automatic tool | %ERD, %ERS, CIT2FS, e-LORETA | • more brain activation regions for VR<br>• higher cognitive load for the VR case<br>• CIT2FS performed better for VR |
| (Dey et al., 2019) [10] | 14 | adaptive target training system | α | 9–13 Hz, notch, >0.3 Hz | threshold of max absolute values or variance, VI | mean of squares, mean of last 4 epochs, TFR, Monte Carlo permutation test | • the brain has the ability to adjust for an increased task load without impacting performance |

| Authors, Year, Reference | Participants | Stimuli | Frequency Bands/ Range | Data Preprocessing | Artifact Removal | Classification Technique/Statistical Analysis | Main findings |
|---|---|---|---|---|---|---|---|
| (Tremmel and Krusienski, 2019) [11] | 15 | n-back task | δ, α, β, γ | 0–58 Hz, Welch's method | WCF EMG-surrogate Regression | LDA, 5-fold cross-validation | • the proposed combination of two artifact suppression techniques reduced artifacts<br>• acceptable workload decoding performance |
| (Tremmel et al., 2019) [12] | 15 | n-back task | θ, α, β, γ, HF | >5 Hz | HF suppression | Welch's method, Spearman's correlation, rLDA, 4-fold cross-validation | • EEG measurements can differentiate between three levels of workload<br>• better classification accuracies using HF suppression, θ and β bands |
| (Sun et al., 2019) [77] | 28 | 2D vs. VR | | 0.05–100 Hz | 100 μV threshold | N1, P2 mean, SD, 3-way rmANOVA, Greenhouse–Geisser correction | • low cognitive load and improved learning in VE for low spatial ability participants<br>• no significant difference in cognitive load and less learning for high spatial ability subjects |
| (Van Goethem et al., 2020) [58] | 8 | 2D vs. 3D shapes | | | | QStates software Paired Sample *t*-test | • no significant difference for the assignment in VR and 2D |
| (Nenna et al., 2020) [76] | 22 | Visual discrimination task | θ, α1, α2, β, γ | BeMoBIL, FIR (0.2–90 Hz), <40 Hz | automated rejection, VI, AMICA | PSD, ANOVA, Mauchly's test, Greenhouse–Geisser correction, BC, P3-SNR | • even with simple tasks, walking affects the simultaneous processing of visual stimuli. |
| (Škola et al., 2020) [13] | 15 | VR storytelling | θ, α, β2 | 1.5–100 Hz, notch | ASR, MARA, AMICA | PSD, neural de/synchronization | • high levels of immersion and engagement<br>• increased cognitive processing |
| (Haruna et al., 2020) [14] | 9 | BCI-Haptic (with/without VR) | θ, α, β | 0.5 s epochs | | ffDTF, SCoT, VAR | • reduced cognitive burden on the remote machine system operator |

| Authors, Year, Reference | Participants | Stimuli | Frequency Bands/ Range | Data Preprocessing | Artifact Removal | Classification Technique/Statistical Analysis | Main findings |
|---|---|---|---|---|---|---|---|
| (Baceviciute et al., 2020) [15] | 78 | text (3 formats) | θ, α | 0.5–100 Hz, notch, PSD (DFT) | ICA (MARA) | mean peak frequency, one-way ANOVA, Tukey's HSD, KWT, BC post hoc Dunn's tests | • reading was superior to listening for retention, self-efficacy, and extraneous attention<br>• text from a virtual book was less cognitively demanding than the overlay interface |
| (Baceviciute et al., 2021) [16] | 48 | Text (book vs. VR) | θ, α, β | 0.5–100 Hz, notch | VI, ICA (MARA), automatic channel rejection | mean PSD, independent samples *t*-test, BC | • participants achieved notably better results on a knowledge transfer test when using VR.<br>• reading in VR was more cognitively effortful and less time efficient |
| (Tian, Zhang, et al., 2021) [17] | 40 | Films (VR vs. 2D) | θ, α, β | FIR, notch, WT | VI, ICA | frequency band energy | • fast-cutting rate brings a greater load |
| (Tian, Wang, et al., 2021) [18] | 30 | films | θ, α, β | notch, 0.5–90 Hz, 0.1–30 Hz, WT | ICA, 100 μV threshold | EEG energy, SVM | • pivotal role of frontal and parietal brain regions on VEPs<br>• all editing methods affect cognitive load and immersion in a VE |
| (Redlinger et al., 2021a) [49] | 20 | N-back memory task | θ, α, β1, β2 | notch, >4 Hz, FFT | EOG, VI | power index, rmANOVA, WSRT | • benefits of an HMD used in cognitive training task |
| (Redlinger et al., 2021b) [47] | 20 | N-back memory task | θ, α, β1, β2 | notch, >4 Hz, FFT | ICA, EOG | power index, rmANOVA, WSRT | • benefits for visual angle of 20°<br>• worse results for increased angles |
| (Redlinger and Shao, 2021) [48] | 12 | Game (2d vs. VR) | θ, α, β1, β2 | notch, >4 Hz, FFT | VI, EOG | power index, WSRT | • increased brain activity VR |

**Table 7.** *Cont.*

| Authors, Year, Reference | Participants | Stimuli | Frequency Bands/ Range | Data Preprocessing | Artifact Removal | Classification Technique/Statistical Analysis | Main findings |
|---|---|---|---|---|---|---|---|
| (Aksoy et al., 2021) [19] | 20 | N-back memory task (VR vs. 2D) | | 0.5–30 Hz | 100 μV threshold, VI | mean amplitude, peak amplitude, peak latency of N1, P1, P3, rmANOVA | • the investigation of visual ERPs in cognitive working memory tasks is experimentally feasible, wearing a VR HMD over an EEG headcap |
| (Atici-Ulusu et al., 2021) [86] | 4 | automobile manufacturing factory | β, γ | β, γ wavelength filtering, notch | BrainVision Analyzer 2 software | EEG graph area, SD, One Sample *t*-test, Mean | • participants using AR had lower cognitive load<br>• easy adaptation to AR technology |
| (Lee et al., 2022) [40] | 15 | 3D objects | θ, α | 0.01–120 Hz, notch | ICA | Welch's method | • participants were more intuitive and relaxed when using Vive Wands<br>• the Leap motion has great potential for providing immersive experience |
| (Tehrani et al., 2022) [41] | 10 | VE construction field | θ, α, β | 0.5–60 Hz | ASR, ICA, VI | WPD, SE, Mann–Whitney U test | • the group working at heights reported experiencing higher levels of fatigue compared to the group working on the ground. |
| (Baceviciute et al., 2022) [20] | 63 | text, auditory, text and auditory | θ, α | 0.5–100 Hz, notch, PSD (DFT) | VI, ICA-MARA | mean peak frequency, SD, ANOVA | • the redundancy effect may not generalize to VR |
| (W. Wang et al., 2022) [84] | 20 | flight simulator 2D vs. MR | | | | P300 amplitude, P300 latency, paired *t*-test, WPD, SE | • improved strategy presented for the display interaction system for future aerospace equipment |

AMICA = Adaptive Mixture ICA, ANOVA = ANalysis Of Variance, ASR = Artifact Subspace Reconstruction, BC = Bonferroni Correction, CAR= Common Average Referencing, DFT = Discrete Fourier Transform, ERD = Event-Related Desynchronization, ERS = Event-Related Synchronization, ICA = Independent Component Analysis, ffDTF = full frequency Directed Transfer Function, FFT = Fast Fourier Transform, FIR = Finite Impulse Response, kNN = k-Nearest Neighbor, LDA = Linear Discriminant Analysis, LDFA = Linear Discriminant Function Analysis, MARA = Multiple Artifacts Rejection Algorithm, PSD = Power Spectral Density, RFE-CBR = Recursive Feature Elimination-Correlation Bias Reduction, rLDA = regularized LDA, rmANOVA = repeated-measures ANOVA, SCoT = Smooth Coherence Transform SD = Standard Deviation, SE = Sample Entropy, SNR = Signal to Noise Ratio, SVM = Support Vector Machine, TFR = Time Frequency Representations, VAR = Vector Auto Regressive, VI = Visual Inspection, WCF= Warp Correlation Filter, WPD = Wavelet Packet Decomposition, WSRT = Wilcoxon signed rank test, WT = Wavelet Transform.

4.2.1. Objectives and Outcomes

Several studies investigated immersive digital environments in comparison with real-world or non-immersive environments and their effect on the participants' cognitive load. Comparative studies ([85,86]) between real-life and AR on an automobile assembly line assessed the cognitive load of four workers. In a similar study [14], nine subjects were asked to manipulate a remote machine system in the VR space, using Brain Computer Interface (BCI). A comparison was made on driving in a VR environment with that in a traditional non-VR environment [39]. An MR flight simulator was proposed in [84] and compared with a 2D display system. Reduced cognitive burden was measured in the experiments using digital environments for all the above studies, providing evidence that digital environments can improve learning and performance results. Another study also comparing a 2D and VR version of a game similar to "Tetris", found significant increase in frontal midline θ rhythm for the Virtual Environment (VE) condition, associated with higher levels of engagement [48].

In contrast with the previously mentioned studies, some articles showed higher workload when participants read text in a VR lab simulation, in relation to a PC-based application [5] and also in comparison with a traditional book [16]. Ambiguous learning outcomes were presented, where the earlier study [5] supports less learning in the VE and the latter article [16] reports better results in transfer tests. The effects of adding narration were also investigated in [5], showing no significant difference. The same text presented as an overlay, embedded in virtual book or as audio was further explored in a later study [15], showing that reading text from a virtual book was less cognitively demanding than from an overlay interface. On the other hand, a study focused on cognitive effort needed for tasks with 3D solids (in VR) and for the same 2D solids printed on paper [58], showed no significant difference on the cognitive load for the assignment in VR and in 2D. Another VR vs. 2D comparative study [77] was conducted on two groups of participants with different levels of spatial ability (low and high). The results showed that low-spatial ability students' cognitive load and learning ability were increased in the VR environment, while for the high-spatial ability subjects no significant change on the cognitive load were found for the two different environments and lower learning performance for the VE.

A satisfactory multi-level workload classification accuracy was achieved for three mental workload levels and distinct development trends between 2D and 3D interfaces were found in a simulated flight experiment [51]. An N-back working memory task was implemented in article [19] in both VR and 2D environments. Findings indicated that it is possible to investigate cognitive workload wearing a VR HMD over an EEG cap.

The majority of the experiments were conducted only within a VR, AR or MR environment, evaluating various aspects of their effects. A combined evaluation of an application mixing interactive virtual reality (VR) experience with 360° storytelling [13] showed increased cognitive processing, which was not associated with overloading. It was considered as evidence of participants' high levels of immersion and engagement. The authors of [17,18] investigated the application of cognitive event segmentation theory on VR films. The results suggest that 2D and VR films with fast-cutting rates are more cognitively demanding and that the frontal and parietal brain regions are the main sources of visual-evoked potentials.

In some papers, the main objective was to investigate how the task complexity can influence cognitive load and performance. A study focused on redundancy [20], where a VR educational application presented information to a total of 73 participants in two different sensory channels at the same time (both written and auditory). The results suggest that redundant content was not found to be more cognitively demanding than written content alone.

An adaptive system, Levity [9], was implemented to measure the user's level of cognitive load and interactively adjust the complexity of a visual search task. In a similar study [10], a cognitively adaptive selected target training system was explored, which required 14 participants to complete 20 levels of increasing complexity. $\alpha$ power decrease [9]

and significant differences in signal power between 1 and 30 Hz [10] were identified when the task complexity increased. A visual discrimination task with two different levels of complexity was designed in [76], while participants were standing (single-task) and while walking overground (dual-task). Even when the task was easy, the effect of walking on the simultaneous processing of visual stimuli was evident. An adaptive training task at three separate visual angles was performed by 20 subjects in study [47]. In contrast with previous studies, the study discovered that enlarging the screen size up to a visual angle of around 20°, led to improved performance in memory and learning tasks. However, beyond this point, further increases in size were associated with a decline in task performance.

Other aspects of the cognitive load assessment were investigated by several studies. A publicly available dataset (VIEW) of visual tasks, where participants were asked to rate multimedia quality levels of images was evaluated by the authors of [89]. The results reflected a large spread of accuracies for the individual cognitive load predictions across subjects. The feasibility of passively monitoring cognitive workloads during classical gesture tasks was evaluated in another research study [40]. The results indicated that participants feel more intuitive and relaxed when using Vive wands in the VE. The objective of [41] was to examine how working at heights contributes to the development of mental fatigue, with the aim of preventing the risk of falls. The results suggested that height had a negative impact on subjects' alertness and indicated higher levels of mental fatigue. The authors of [11,12] implemented EMG artifact removal techniques (WCF, EMG-surrogate Regression, HF suppression), which led to better classification accuracies.

### 4.2.2. Data Preprocessing and Artifact Removal

The majority of researchers [9,10,13,15–20,40,41,47–49,51,85,86] have applied a notch filter in order to remove the powerline noise. A variety of band-pass filters were also applied: 9–13 Hz [10]; 0.5–40 Hz [51]; 0.5–30 Hz [19]; 0.05–30 Hz [77]; 0–58 Hz [11]; 0.5–60 Hz [41]; 0.1–90 and 0.1–30 Hz [18]; 0.2–90 Hz [76]; 0.5–100 Hz [20]; 1.5–100 Hz [13]; 0.01–120 Hz [40]. Only high-pass filters of 0.3 Hz [10]; 4 Hz [47–49] and 5 Hz [12] were used, while the authors of [9] report the application of a low pass filter. In several papers, Power Spectral Density (PSD) estimates were constructed using Discrete Fourier Transform (DFT) [15,16,20] and Fast Fourier Transform (FFT) [47–49].

A visual inspection was also performed in [10,16,17,48,49,76] to reject low-quality trials. The Welch's method ([11,12]), a conservative Hampel filter [12], mean peak frequency estimates [20], and Wavelet Transformation (WT) [17] were applied in a few studies. Automated methods, such as QStates software [58] and the BeMoBIL Preprocessing Pipeline [76] were also selected for the preprocessing stage. A significant number of authors performed Independent Component Analysis (ICA) [15–18,20,39–41,47,49,51]. A Visual Inspection (VI) to manually clean artifacts was also implemented in several papers [8,10,20,41,48]. Muscle artifacts were identified from comparison with the EOG data in a few cases [47–49]. Artifacts were automatically removed using BrainVision Analyzer 2 software [85,86]; ABM's proprietary software [5]; Artifact Subspace Reconstruction algorithm (ASR) [13,41]; Multiple-Artifact Rejection Algorithm (MARA) [13,20]; a combination of two methods [11]: Warp Correlation Filter (WCF) and EMG-surrogate Regression; Adaptive Mixture Independent Component Analysis (AMICA) and a low-pass filter at 40 Hz [76]; 100 μV threshold [18,76]; Common Average Referencing (CAR) and Elliptical BPF [39] and EEGLAB automatic channel rejection tool [20].

### 4.2.3. Signal Analysis

The EEG signal was not decomposed into frequency bands in a few studies [5,19]. The vast majority of authors [9–12,14–18,20,40,41,51,85,86,89] considered various subsets of the δ, θ, α, β and γ frequency bands. Meanwhile, for the articles [8,13,39,47–49], the bands of interest also include the α and/or β sub-bands. Higher frequency range, termed HF (70–100 Hz), was additionally analyzed in study [12].

#### 4.2.4. Statistical Analysis

A variety of statistical analysis methods were implemented on EEG data for the assessment of cognitive load. According to another approach, the area under the average EEG graph was investigated, where higher cognitive load was considered when the absolute value of the area was lower [85,86]. One Sample *t*-test was used for evaluation. The authors of [9] measured the center of gravity frequency on the shape of the $\alpha$ peak and event-related desynchronization (ERD) as the percentage of $\alpha$ band power decreased. Another method was implemented in [10], based on the calculation of the mean of squared signals, Time-Frequency Analysis (TFR) for the hardest and easiest levels of the training task and Monte Carlo cluster-based permutation testing. In a similar direction, the sum of squares of all points in each frequency band were calculated in [17] to represent the energy of the signal.

PSD estimates per frequency band were compared between the three experimental conditions using one-way ANOVAs in study [15] and post hoc Tukey's Honest Significant Difference (HSD) tests, nonparametric Kruskal–Wallis test (KWT) and Bonferroni-corrected (BC) post hoc Dunn's tests were used for the analysis of the self-reported data. The statistical analysis of another article [16] consisted of the computation of mean PSD, mean peak frequency estimates and independent samples (IS) *t*-test. The researchers of [20], focused on the between-group differences in mean EEG power, using one-way ANOVAs and BC. For the P3 Signal-to-Noise Ratio (SNR) analysis presented in [76], PSDs, ANOVAs, the Mauchly's test of sphericity, the Greenhouse–Geisser correction and the Bonferroni Correction (BC) method were used. The authors of [48] calculated the signal band power and statistically analyzed it with Wilcoxon signed-rank tests, while the authors of [58], used the software provided by the EEG hardware company to assess the cognitive load.

Statistical analysis of [47,49] included the calculation of the sum of power values, mean and standard deviation, with repeated measures ANOVA (rmANOVA), and a nonparametric, Wilcoxon Signed Rank Test (WSRT). In study [19], Event-Related Potential (ERP) analysis was conducted, where mean amplitude, peak amplitude, peak latency of N1, P1, P3 were computed and subsequently they were analyzed with rmANOVA. A similar statistical ERP analysis was presented in article [77], where N1 and P2 mean and SD amplitude were evaluated through 3-way rmANOVAs and Greenhouse–Geisser correction.

The information flow was computed in [14] with the use of smooth coherence transform (SCoT) library, full frequency Directed Transfer Function (ffDTF) and stationary vector autoregressive (VAR) model of the 30th order. To assess the participants' cognitive load, the authors of [13] used PSDs and calculated the index of neural de/synchronization as the percentage change in the absolute band power between the baseline and the final phase. The P300 average amplitudes and the average latency were analyzed by the authors of [84] and paired *t*-test was applied. For the assessment of mental fatigue levels Wavelet Packet Decomposition (WPD) and Sample entropy (SE) were computed, and seven indices for measuring mental fatigue were obtained, namely $\theta$, $\alpha$, $\beta$, $\alpha/\beta$, $\theta/\alpha$, $\theta/\beta$, and $(\theta + \alpha)/\beta$, A Mann–Whitney U test was performed to statistically compare mean PSD values.

#### 4.2.5. Classification Methods

In a limited number of articles, the cognitive load was assessed using classification methods. The authors of [5] developed a Linear Discriminant Function Analysis (LDFA) classifier, they also derived the absolute and relative power spectra variables using stepwise regression and calculated ANOVAs for the PC and VR condition. The authors of [18] studied the Visual Evoked Potentials (VEP) and a Support Vector Machine (SVM) classifier was implemented on EEG energy features, sLORETA and ANOVA. In article [89], four position-dependent ratios were utilized as their main evaluation criteria (the average $\alpha/\beta$ ratio, the average ratio of $\alpha$ and $\beta$, the variability of $\alpha$ and $\theta$, and the variability of $\alpha$ and $\beta$) using the k-Nearest-Neighbor (KNN) classification and regression approach to evaluate whether the ratio levels can be correctly employed to identify the different cognitive load levels.

The authors of [39] extracted EEG features from the percentage ERD, followed by the percentage ERS. A CIT2FS-induced pattern classifier, based on source localization through e-LORETA, was implemented to classify into three cognitive load classes: High, Medium and Low. In a similar direction, exact Low-Resolution Electromagnetic Tomography (eLORETA) and a validated atlas (AAL-116) were employed by [51], resulting in 80 ROIs. PLI networks were calculated for each epoch and frequency band. Global efficiency (EG) and local efficiency (EL) of brain networks were employed, feature extraction was based on RFE-CBR, and classification was performed with Linear Discriminant Analysis (LDA) using 100 repetitions of 10-fold cross-validation. Moreover, a permutation test and two-way Analysis of Variance (ANOVA) were performed. An LDA classifier with 5-fold cross-validation was used by [11], while in [13] the Spearman's correlation was computed and an rLDA classifier was implemented with a 4-fold cross-validation. The authors of [40] used PSD-based feature extraction and Welch's method classification for the cognitive load estimation.

### 4.3. Immersion

This category covers studies related to body ownership, sense of presence (oneness), agency and engagement. Body ownership is associated with the illusory perception that an artificial or virtual object becomes part of one's body. Sense of presence (or oneness) refers to the situation where a person immersed in a VE gets the feeling that it is a real-life environment. Agency is related to the sense of a person's ability to influence a digital environment. Table 8 presents a summary of findings in the field of immersion.

#### 4.3.1. Objectives and Outcomes

One of the most commonly used experiments in the field of body ownership is the rubber hand illusion, where participants perceive a model of a hand as an integral part of their own body. A realistic 3D representation of a hand was chosen to create the same illusion in both VR and AR in [88]. The findings indicate that $\beta$ and $\gamma$ bands in premotor cortex activity are associated with body ownership statements. In another experiment [22], the participant's mirror-neuron system (MNS) and the error-monitoring system (EMS) were stimulated by involuntary and unexpected virtual hand bounces. The results indicated higher Pe/P300 effect among participants who had a stronger rubber hand illusion experience and increased tendency for affective empathy. Agency and responsibility were also investigated in [50] by studying the control of movements of an embodied avatar, via BCI technology in three experimental conditions: no VR, Steady-State Visually Evoked Potentials (SSVEP) and observation with VR. Evidence was found that the sense of agency in BCI systems is strongest for sensorimotor areas activation.

Differences between 2D and VR video watching from three categories (sports, news, and advertisements) showed higher $\beta$-wave activity for VR compared to 2D [65]. It was also shown that videos with fast transitions induced higher $\beta$ activity. The optimum exposure duration to a virtual classroom environment was evaluated in [21]. The results suggested significant differences in brain activity between realistic and non-realistic VEs. Furthermore, it was found that the time required by the brain to perceive and adapt to the artificial environment is at least 42.8 s. The authors of [74] showed the feasibility of adding irrelevant auditory stimuli in experiments, for the evaluation of the levels of immersion in both 2D and 3D environments in tasks with increasing difficulty. Another study in the field of education explored the effects of a VR training application on immersion, in comparison with a lecture-based design [23]. The results indicated increased attention-related and meditation-related brain wave activity and desynchronized $\alpha$ waves in the VR environment.

A new method was proposed by the authors of [14] to measure sense of oneness in a visual haptics experiment. A comparative study of cycling in an immersive and non-immersive VE was conducted in [59]. The results confirmed that the participants were more engaged and performed better in the immersive VE. Ethnic bias in empathic resonance to pain was investigated in [60]. Amplified beta ERD was measured when a digital agent of the same color (with the participant's virtual body) "experienced" pain.

**Table 8.** Immersion.

| Authors, Year, Reference | Participants | Objective | Stimuli | Frequency Bands/ Range | Data Preprocessing | Artifact Removal | Classification Technique/Statistical Analysis | Main Findings |
|---|---|---|---|---|---|---|---|---|
| (Burns and Fairclough, 2015) [74] | 20 | immersion | auditory oddball task (2D/VR) | | 0.1–30 Hz | | GA ERP, mean amplitudes | • immersion was characterized as the focused attention on external auditory stimuli unrelated to the game, and it was assessed indirectly by analyzing ERPs in response to an auditory oddball task |
| (Škola and Liarokapis, 2016) [88] | 30 | body ownership | virtual hand (Physical, VR, AR) | δ, θ, α, β, γ | 1.5–95 Hz, notch, FFT | ICA (MARA) | PC | • correlation between ownership statements and β, γ bands in premotor cortex activity |
| (Baka et al., 2018) [21] | 33 | sense of presence | VE realistic, non-realistic | θ, α, β1, β2 | 0.1–60 Hz, notch, FFT, 10 ROI | VI | Mann–Whitney, KWT | • non-realistic and realistic VEs induce different brain oscillations<br>• the duration of VR applications must be at least 42.8 s in order to be effective |
| (Kweon et al., 2018) [65] | 20 | immersion | videos (2D/VR) | α, β | | | paired *t*-test, α, β wave difference 2D/VR | • higher brain activity and enhanced experience in the VR condition<br>• differences in video genre |
| (Haruna et al., 2020) [14] | 9 | sense of oneness | visual haptics feedback | θ, α, β | 0.5 s epochs | | ffDTF, SCoT, VAR | • the proposed method can measure effectively participants' sense of presence |
| (Raz et al., 2020) [22] | 18 | body ownership | virtual hand | mu rhythm | >0.1 Hz, Morlet WT | ICA, VI | ERSP, ERP, cluster-based permutation, PC, ANOVA, two-tailed signed rank test | • The alternative body can induce sensorimotor sensitivity through synchronicity and semantic congruence.<br>• no association was found between Mu power and Pe/P300 |

**Table 8.** *Cont.*

| Authors, Year, Reference | Participants | Objective | Stimuli | Frequency Bands/ Range | Data Preprocessing | Artifact Removal | Classification Technique/Statistical Analysis | Main Findings |
|---|---|---|---|---|---|---|---|---|
| (Nierula et al., 2021) [50] | 29 | body ownership, agency | BCI (no VR/VR) | α | 0.5–40 Hz, notch, CSP, HT | VI, VMA, MD, ICA | sBEM, ERD%, Tikhonov-regularized minimum-norm | • the sense of agency can be generated by controlling movements with BCI<br>• agency and responsibility are correlated with increased activity of sensorimotor areas |
| (Bogacz et al., 2021) [59] | 14 | engagement | VE cycling | α | 1–20 Hz, Welch's method | VI | ROI analysis, α power peak | • the subjects were more engaged in the immersive condition |
| (Harjunen et al., 2022) [60] | 58 | embodiment | VR hands, VR agents | β | CSD, FFT | ICA | ERD/ERS, average β ERD, rmANOVA, F-tests, type-III sum of squares, planned pairwise comparisons | • sensorimotor resonance was modulated by changes in bodily resemblance to others' perceived pain |
| (Y.-Y. Wang et al., 2022) [23] | 72 | immersion | images, game | θ, α, β, γ | | | average, log values, MANOVA, MANCOVA | • VR leads to increased brain wave activity related to attention and meditation, as well as desynchronization of α waves |

ANOVA = ANalysis Of Variance, CSP = Common Spatial Pattern, ERD = Event-Related Desynchronization, ERP = Event-Related Potentials, ERS = Event-Related Synchronization, ERSP = Event-Related Spectral Dynamics, ICA = Independent Component Analysis, ffDTF = full frequency Directed Transfer Function, FFT = Fast Fourier Transform, GA = Grand Average, HT = Hilbert Transform, KWT = Kruskal–Wallis test, MANOVA = Multivariate Analysis of Variance, MANCOVA = Multivariate Analysis of Covariance, MARA = Multiple Artifacts Rejection Algorithm, MD = Mahalanobis Distance, PC = Pearson Correlation, ROI = Regions Of Interest, sBEM = symmetric Boundary Element Method, CoT= Smooth Coherence Transform, VAR = Vector AutoRegressive, VI = Visual Inspection, VMA = Variance of the Maximal Activity, WT = Wavelet Transform.

### 4.3.2. Data Pre-Processing and Artifact Removal

A variety of bandpass filters were applied by the researchers as follows: 1–20 Hz [59], 0.1–30 [74], 0.5–40 Hz [50], 0.1–60 Hz [21] and 1.5–95 Hz [88]. The authors of [22] used a 0.1 Hz high-pass filter. A notch filter was applied in [21,88]. A number of other pre-processing methods were also used: FFT [21,60,88], Hilbert Transform (HT) [50], Morlet wavelet [22] and Welch's method [59].

Artifacts were identified by visual inspection in [21,22,50,59]. ICA was also an artifact rejection method performed by a significant number of authors [50,60,65,88]. Several other methods were used as follows: the Variance of the Maximal Activity (VMA) and the Mahalanobis distance (MD) [50], Artifact Subspace Reconstruction (ASR) [65], ICLabel [65].

### 4.3.3. Signal Analysis

All frequency bands, $\delta$, $\theta$, $\alpha$, $\beta$ and $\gamma$ were considered by [88], while only $\alpha$ band was considered in [50,59]. Only $\beta$ band was used in [60]. Brain rhythms of interest included sub-bands only in [21] ($\theta$, $\alpha$, $\beta$ and $\gamma$1). The mu rhythm, defined as the frequency band between 8 and 13 Hz and measured at central electrodes, was investigated in [22].

### 4.3.4. Statistical Analysis

A wide spread of statistical analysis methods on EEG recordings was implemented. The Grand Average (GA) and mean amplitude of ERPs [74]; Pearson Correlation (PC) analysis [22,88]; $\alpha$ and $\beta$ brain activity and paired *t*-tests [65]; the power spectra and the average across segments for ten ROIs, non-parametric tests (Mann–Whitney and KWT) [21]; Event-Related Spectral Perturbations (ERSP), a non-parametric cluster-based permutation test, ANOVAs, FDR-corrected two-sided signed rank test and permutation test [22]; the time points with lowest ERD% extracted with the HT, Common Spatial Pattern (CSP) analysis, the mean, the symmetric Boundary Element Method (sBEM) and the Tikhonov-regularized minimum-norm [50]; the average $\beta$ ERD/ERS for two ROIs (left/right hemisphere), rmANOVA, uANOVA, F-tests with type-III sum of squares and planned pairwise comparisons [60]; differences in peak $\alpha$ power under non-immersive and immersive scenarios, average, log value computation, Multivariate Analysis Of Variance (MANOVA) and Multivariate Analysis of Covariance (MANCOVA) [59].

### 4.4. Spatial Awareness

Spatial navigation wayfinding behavior refers to the ability of a person to find their way to a goal location. Table 9 presents a summary of findings in the field of spatial awareness.

### 4.4.1. Objectives and Outcomes

The authors of [68] suggested that experiments developed for research on spatial navigation should not be static, as kinesthetic and vestibular information may alter $\alpha$ brain waves.

A virtual house environment model was proposed in [52], where the effect of different colors of the target (bathroom door) was studied in association with the participants' age. The results showed no significant differences in brain activity between the different colors of the target door for the young group, while in the elderly group green and red colors evoked a significantly larger P3b with respect to the other door colors. In a similar direction, the effects of three different interior designs (color, graphics, architectural features) were evaluated in [43]. Improvements in some orientation behaviors for the most extensive wayfinding design were found with no significant improvements on performance or in self-reported cognitive state. The assessment of the effects of social characteristics (gender, age, level of education) and the height of the ceiling indicated that wayfinding behavior was influenced by these factors [71].

**Table 9.** Spatial awareness.

| Authors, Year, Reference | Participants | Stimuli | Frequency Bands/ Range | Data Preprocessing | Artifact Removal | Classification Technique/Statistical Analysis | Main Findings |
|---|---|---|---|---|---|---|---|
| (Ehinger et al., 2014) [68] | 5 | triangle completion task | α | 1–120 Hz | VI, AMICA, BEM | ERSP, PCA, k-means, ROI analysis (Monte Carlo) | • Significant differences in brain activity between experiments with static and walking subjects |
| (de Tommaso M et al., 2016) [52] | 28 | VE home colors | 0.5–80 Hz | 0.5–80 Hz | VI, ASA-ANT software, ICA | GA, P3b amplitude and latency, one-way ANOVA, MANOVA, scalp maps, BC | • in young group increase in P3b amplitude for the target, regardless of the door color <br> • in elderly group significantly different P3b amplitudes for green and red target, no increase for the white stimulus |
| (Sharma et al., 2017) [53] | 30 | maze | θ | 4–8 Hz | VI, ICA | %θ change, ERD/ERS, DFT, IS *t*-tests, ANOVA, ROI analysis | • better performance when landmarks where present compared to the no landmark condition |
| (Erkan, 2018) [71] | 340 | maze | θ, α, β | EEG-Analyzer Tool, FFT | Gratton, Coles, and Donchin algorithm | θ, α, β activity | • wayfinding behavior is influenced by personal and social characteristics of people and is related to space height |
| (Gehrke and Gramann, 2021) [54] | 29 | Maze | θ, α | 124–500 Hz | VI, ICA, AMICA | MAD, SD, MD, BEM, k-means clustering, LME, Tukey's, Spectral maps, ERSP | • α power oscillations can be used for the investigation of wayfinding behavior |
| (C.-S. Yang et al., 2021) [24] | 41 | spatial task | α, β | 1–45 Hz | VI, ASR, AMICA | k-means, ERSP, correlation analysis | • participants' preferred spatial reference frame (allocentric/egocentric) may change for different environments |
| (Liang et al., 2021) [25] | 19 | teleporter | δ, θ, α, β | 1–50 Hz, Morlet WT | ASR, ICA | mean, WSRT, SVM | • spatial distance and temporal durations during navigation are associated with different power changes in brain signals |

**Table 9.** *Cont.*

| Authors, Year, Reference | Participants | Stimuli | Frequency Bands/ Range | Data Preprocessing | Artifact Removal | Classification Technique/Statistical Analysis | Main Findings |
|---|---|---|---|---|---|---|---|
| (Ellena et al., 2021) [61] | 22 | avatar | 0.5–30 Hz | 0.5–30 Hz | voltage threshold, SD, ICA | N1 mean amplitudes, rmANOVA, Newman–Keuls | • intrusion of fearful faces into the peripersonal space may heighten the expectation of a visual event occurring in the periphery. |
| (Yi et al., 2022) [78] | 19 | Open dataset | δ, θ, α, β | 1–50 Hz, Morlet WT | ASR, ICA | pMFLR, PCA, cross-validation | • the proposed algorithm leads to an interpretable classification • frontal and parietal δ-θ are the most related to distance judgment oscillations |
| (Zhu et al., 2022) [42] | 30 | VE hospital | θ, α, β | 1–50 Hz, PREP Pipeline, SSI, CSP | ASR, ICA, VI | log transform, RF, 5-fold cross-validation ROC | • navigational uncertainty state can be potentially identified by EEG data processing |
| (Kalantari et al., 2022) [43] | 63 | VE hospital | δ, θ, α, β, γ | | ICA | IC cluster analysis, one-way ANOVA, post hoc Tukey HSD, ERSP | • improvements in orientation behaviors for the extensive wayfinding design and greater neurological activation • no improvement in wayfinding performance and self-reported experience |

AMICA = Adaptive Mixture, ANOVA = Analysis Of Variance, ASR = Artifact Subspace Reconstruction, BEM = Boundary Element Method, BC = Bonferroni Correction, DFT = Discrete Fourier Transform, ERSP = Event-Related Spectral Dynamics, GA = Grand Average, IC = Independent Component, ICA = Independent Component Analysis, IS = Independent Samples, HSD = Honest Significant Difference, LME = Linear Mixed Effects, MANOVA = Multivariate Analysis of Variance, MAD = Mean Absolute Distance, MD = Mahalanobis Distance, PCA = Principal Component Analysis, pMFLR = penalized Multiple Functional Logistic Regression, RF = Random Forest, rmANOVA = repeated-measures ANOVA, ROC = Receiver Operating Characteristics, ROI = Regions Of Interest, SD = Standard Deviation, SVM = Support Vector Machine, VI = Visual Inspection, WSRT = Wilcoxon signed rank test, WT = Wavelet Transform.

Maze-like environments were designed in [53] to investigate the influence of landmarks on performance during navigation. The findings revealed better performance (fewer errors, shorter times to complete the task) when landmarks were present compared to the no landmark condition mainly on the left-hemispheric region. A virtual maze was also used for the assessment of spatial learning [54]. A significant $\alpha$ power decrease was identified in the presence of novel walls and with the increase in walking distance. The possibility of using EEG as a metric for the identification of uncertainty states, while navigating in a VR hospital environment, was the research objective of a study [42], where high accuracies were achieved with the use of the proposed machine-learning classification approach.

A three-stage algorithm [78] was applied to an openly available dataset, leading to interpretable classification and indicating a pivotal role of frontal and parietal delta-theta oscillations in distance judgment. In a similar direction, in paper [25], a teleportation task was implemented to assess perceived distance and duration. Results suggested that occipital $\alpha$ frequencies are associated with both distance and duration, but for the rest of the brain regions and frequency bands no common brain activity could be linked with both distance and duration. Another study [61] focused on the participants' modulation on spatial perspective when fearful faces appear in close proximity, indicating a potential threat. The results confirmed a reduction in N1 mean amplitude, proportional to the speed of their reaction and elicited by the peripheral probe for near fearful relative to neutral faces. Allocentric and egocentric navigation was the research objective of [24], where both behavioral and brain dynamics results indicated alterations in subjects' spatial reference frame, depending on the environment type.

### 4.4.2. Data Preprocessing and Artifact Removal

A variety of band-pass filters was used by the researchers as follows: 0.5–30 Hz [61]; 4–8 Hz [53], 124–500 Hz [54]; 1–45 Hz [24]; 1–50 Hz [78]; 0.1–50 Hz [25,42], 1–120 Hz in study [68]. A number of other methods were also used: the PREP Pipeline for the removal of line noise [42]; the 'EEG-Analyzer Tool', developed by the authors of [71] for creation of 'power spectra' graphics; and Morlet WT [25].

Artifacts were identified by visual inspection in [24,42,52,53,68]. ICA was also an artifact rejection method preferred by a significant number of authors: [25,42,43,52–54,61,78]. Several other methods were used as follows: the algorithm developed by Gratton, Coles, and Donchin [71], AMICA [24,54,68]; BEM [54,68]; ASR [24,25,42,78]; the ASA-ANT software [52]; CSP [42]; the removal of flat line channels [42]; a voltage threshold of 400 µV, SD [61].

### 4.4.3. Signal Analysis

The main frequency bands ($\delta$, $\theta$, $\alpha$, $\beta$, $\gamma$) or a subset of them were used in almost all papers [24,42,43,53,54,68,71,78], where different frequency ranges were considered rather than frequency bands by the authors of [52,61], while there were not any articles reposting the use of sub-bands.

### 4.4.4. Statistical Analysis

The percentage change and the ratio ERD/ERS in $\theta$ power were computed in [53] and DFT was computed for eight ROIs. Independent sample $t$-tests and univariate ANOVA were applied to the behavioral measures. The P3b amplitude and latencies were estimated in [52], by one-way ANOVA, MANOVA and a post hoc BC. Scalp Maps of the Grand Average (GA) of the P3b and Statistical Probability Maps (SPM), were constructed and evaluated. The activity of the frequency bands of interest ($\theta$, $\alpha$, $\beta$) was used as a metric by the authors of [71]. A number of statistical methods were used in [54], including Mean Absolute Distance (MAD), SD, Mahalanobis Distance (MD), BEM, k-means clustering, Linear Mixed Effects (LME), Tukey's, Spectral maps and ERSPs. According to another approach [61], the first maximal negative deflection after T1 was used for the evaluation of the N1 component of the left and right temporo-occipital recording sites. Subsequently, rmANOVA and the Newman–Keuls test were used.

4.4.5. Classification Methods

Several analysis approaches were demonstrated in [54]. Mean absolute amplitude, SD, the MD, a dipole fitting procedure and an ROI-driven repetitive k-means clustering approach were employed. EEG analysis was based on spectral maps and ERSPs. For the statistical analysis of the EEG signals, linear mixed effects model and Tukey's correction were evaluated. Significant modulation of α oscillations was identified through single-trial regression. The same clustering method, k-means, along with ERSP and correlation analysis for four ROIs (occipital, frontal, parietal and central) were used by the authors of [24]. Moreover, in study [68], k-means clustering was applied, together with ERSPs using a Morlet Wavelet transformation, Principal Component Analysis (PCA) with an ROI analysis with Monte Carlo tests.

An SVM classifier was implemented in [25], after the computation of mean band power and Wilcoxon signed rank tests were run for the statistical analysis on the EEG recordings. Penalized Multiple Functional Logistic Regression (pMFLR), PCA, and cross-validation were used to classify human behaviors in study [78]. A Random Forest (RF) classifier with 100 trees was applied in [42]. A 5-fold cross-validation and the area under the Receiver Operating Characteristics (ROC) curve were additionally computed. IC cluster analysis was applied in [43], localized to Brodmann Area 18 (BA18) of the brain. One-way ANOVA and the post hoc Tukey HSD method for multiple comparisons were also applied. Finally, ERSPs were calculated and tested via the bootstrap re-sampling method.

*4.5. Interaction with the Digital Environment*

In this section a collection of studies is included, investigating different aspects of the effect that a VE may have on human cognition. Some of the objectives being evaluated are the following: cognitive conflict, performance, human communication/collaboration and affordance. It should be noted that the description below includes only the articles that measured performance and work efficiency through EEG signal analysis, and not by other measurements, such as time to complete the task, etc. Table 10 presents a summary of findings in the field of interaction with the digital environment.

4.5.1. Objectives and Outcomes

Studying the impact of transitional affordances, it was found that perception is influenced by potential actions afforded by an environment [81]. Cognitive conflict occurs when there is a mismatch between the perceived and the expected results of one's action. The authors of [27] designed a VE to study cognitive conflict for three hand representations (realistic, robotic, arrow). ERP analysis showed that participants were more sensitive in cognitive conflict occurrences for the more realistic representations. A model (BCINet) was proposed and tested on two datasets (CC and pHRC) in comparison with established models (EEGNet, DeepConvNet, ShallowNet) [29]. The results showed significantly higher classification accuracy with less trainable parameters. The same authors also evaluated the impact of task duration on the assessment of cognitive conflict, where more pronounced brain activity was found in tasks with longer completion time [30]. In a later study [31], the authors investigated how the velocity of hand movements impacts human brain responses using an object selection task. According to their findings, the integration of hand movements with visual and proprioceptive information during interactions with real and virtual objects requires velocity as an essential component.

**Table 10.** Interaction with the digital environment.

| Authors, Year, Reference | Participants | Objective | Stimuli | Frequency Bands/ Range | Data Preprocessing | Artifact Removal | Classification Technique/Statistical Analysis | Main Findings |
|---|---|---|---|---|---|---|---|---|
| (Hubbard et al., 2017) [26] | 12 | learning performance | Working memory task | α, β2 | FFT | | ERP, TFR | • learning performance can be improved using a VR and EEG-feedback system |
| (Singh et al., 2018) [27] | 32 | cognitive conflict | object selection task | | 0.5–50 Hz | VI | PEN, P300, rmANCOVA, mmANOVA | • close-to-real VEs induce higher brain activity in cognitive conflict tasks |
| (Tromp et al., 2018) [69] | 20 | language comprehension | VE restaurant | 0.01–40 Hz | 0.01–40 Hz | Brain Vision Analyzer | ERPs, N400, ANOVA, Greenhouse-Geisser correction | • a N400 effect was observed when the mismatch mechanisms were activated |
| (Spapé et al., 2019) [55] | 66 | message meaning | game | | 0.2–80 Hz, notch, <40 Hz, | ICA, VI, Autoreject algorithm | rmANOVA, N1, MFN, P3, LPP | • the decoding of a message precedes the processing of the message source (i.e., the messenger) |
| (Djebbara et al., 2019) [81] | 19 | transitional affordance | VE Go/No Go | 0.2–40 Hz | 1–100 Hz | ICA, VI, SD | VEP, MRCP, Peak Analysis, rmANOVA, Tukey's HSD | • potential actions afforded by an environment, may alter perception |
| (J. Li et al., 2020) [79] | 30 | work efficiency | 3 VEs/lighting | β | | | PC | • the efficiency of human work is primarily associated with the right temporal lobe region and the β rhythms. |
| (Foerster et al., 2020) [28] | 40 | motor learning | labeled novel tools | β | 1–50 Hz, notch, | voltage thresholds | ERD/ERS, pairwise comparison, two-tailed *t*-tests, cluster analysis (Monte Carlo) | • language modulated the learning of novel tool use • β power was reduced while using labeled tools |

**Table 10.** *Cont.*

| Authors, Year, Reference | Participants | Objective | Stimuli | Frequency Bands/ Range | Data Preprocessing | Artifact Removal | Classification Technique/Statistical Analysis | Main Findings |
|---|---|---|---|---|---|---|---|---|
| (Choi et al., 2020) [67] | 14 | performance, presence | BCI | 8–36 Hz | data augmentation | | FBCSP, LDA, ANOVA, 4-fold cross-validation, Mann–Whitney U test, BC, ERD ratio | • embodiable feedback induces higher control performance, more distinctive brain activity patterns and enhanced cortical activation |
| (Singh and Tao, 2020) [29] | 26 | cognitive conflict | CC, pHRC datasets | | | | BCINet, EEGNet, DeepConvNet, ShallowNet | • the proposed BCINet model has better classification performance than other well-known models |
| (Singh et al., 2020) [30] | 33 | cognitive conflict | object selection task | | 0.5–50 Hz | ICA, VI | PEN, Pe, PC, rmANOVA | • longer object selection tasks are more informative for cognitive conflict evaluation |
| (Singh et al., 2021) [31] | 20 | cognitive conflict | object selection task | δ, θ, α, β | 0.1–40 Hz | Kurtosis, ICA, DIPFIT, BESA | PEN, Pe, ANOVA, ANCOVA, One-sample *t*-tests, 1000-fold permutation test | • velocity is an important factor for combining hand movements with visual and proprioceptive data while interacting with real or virtual objects |
| (Immink et al., 2021) [44] | 45 | performance | game marksmanship | | 0.1–40 Hz, IRASA | ICA, >150 µV, flat channels, EMG, ECG, EOG | REML, Type II Wald χ2-tests | • the 1/f aperiodic parameters are the most correlated parameters with predicting visuomotor performance |

**Table 10.** *Cont.*

| Authors, Year, Reference | Participants | Objective | Stimuli | Frequency Bands/ Range | Data Preprocessing | Artifact Removal | Classification Technique/Statistical Analysis | Main Findings |
|---|---|---|---|---|---|---|---|---|
| (Foerster and Goslin, 2021) [32] | 37 | affordance | virtual objects | θ, α, β, mu band | 0.1–40 Hz, Laplacian filter, FFT, TFR | Autoreject algorithm, frontal and prefrontal exclusion | ITC, rmANOVA | • the results endorse the embodied cognition approach as opposed to the reasoning-based approach for object processing |
| (Yu et al., 2021) [80] | 36 | reorganizations of functional brain networks | 2D, 3D videos | α, β, γ | | | SVM, RF | • the proposed classifier (SVM) can be utilized to study the neural mechanisms underlying various visual experiences from the perspective of a brain network, with an accuracy of 0.908 |
| (Gumilar et al., 2021) [72] | 24 | inter-brain synchrony | real world vs. VR Avatar | δ, θ, α, β, γ | 0.5–60 Hz, notch, automated pipeline | VI, ICA | eLORETA, PLV | • VR induces similar with the real-world inter-brain synchrony |
| (Cruz-Garza et al., 2022) [62] | 23 | performance | VE classroom | δ, θ, α, β, γ | 0.5–50 Hz, frequency band-power, PDC | ASR, ICLabel, ICA | KWT, k-SVM | • no significant differences on performance • significant changes in EEG features for the short-term memory tasks |
| (Y.-Y. Wang et al., 2022) [23] | 72 | creativity | | θ, α, β, γ | | | MANOVA, MANCOVA | • positive VR impact on the feasibility of the creative process • No significant effects of VR on variety and novelty |

**Table 10.** *Cont.*

| Authors, Year, Reference | Participants | Objective | Stimuli | Frequency Bands/ Range | Data Preprocessing | Artifact Removal | Classification Technique/Statistical Analysis | Main Findings |
|---|---|---|---|---|---|---|---|---|
| (Gregory et al., 2022) [64] | 49 | Working memory performance | Memory task (Social/non-social cue) | θ, α | 0.5–36 Hz | VI, ICA | TFR (Morlet WT) | • social cue altered working memory for status information, but did not affect location information • working memory for both status and location information was influenced by non-social cues. |
| (Giannopulu et al., 2022) [83] | 27 | mental imagery, creativity | virtual objects | β, γ | 1–80 Hz, PSD, PDC | VI, ICA | Levene's test, paired sample *t*-tests, rmANOVA, PC, PCA, Factor Analysis, Bartlett's test | • creativity assessment cannot be limited to a single brain area, but it should investigate various interconnected networks |

ANOVA = Analysis Of Variance, ANCOVA = Analysis of Covariance, ASR = Artifact Subspace Reconstruction, BC = Bonferroni-correction, eLORETA = exact Low-Resolution Electromagnetic Tomography, ERD = Event-Related Desynchronization, ERP = Event-Related Potentials, ERS = Event-Related Synchronization, FBCSP= Filter Bank Common Spatial Pattern, FFT = Fast Fourier Transform, HSD = Honest Significant Difference, ICA = Independent Component Analysis, IRASA = Irregular-Resampling Auto-Spectral Analysis, ITC = Inter-Trial Coherence, KWT = Kruskal–Wallis test, LDA = Linear Discriminant Analysis, LPP = Late Positive Potential, MANOVA = Multivariate Analysis of Variance, MANCOVA = Multivariate Analysis of Covariance, MFN = Medial Frontal Negativity, MMANOVA = Multilevel MANOVA, MRCP = Movement-Related Cortical Potentials, PC = Pearson Correlation, PCA = Principal Component Analysis, PDC = Partial Directed Coherence, PEN = Prediction Error Negativity, PLV = Phase Locking Value, PSD = Power Spectral Density, REML = Restricted Maximum Likelihood, RF = Random Forest, rmANOVA = repeated-measures ANOVA, SVM = Support Vector Machine, TFR = Time Frequency Representations, VEP = Visual Evoked Potentials, VI = Visual Inspection, WT = Wavelet Transform.

The objective of several studies was performance and work efficiency. An adaptive to the users' cognitive state system was designed by the authors of [26], aiming to enhance learning. The results indicated the feasibility of this system and also the need for efficient artifact removal and feature selection methods to improve prediction accuracy. A total of three VEs (open natural, semi-open library, and closed basement) were designed in [79] to investigate the influence of natural light on work efficiency. The findings indicated a strong correlation between the β rhythms of the right temporal lobe region of the brain and efficiency. The findings of [44] indicated that individual 1/f intercept and slope parameters of aperiodic resting state neural activity could be used in predicting visuomotor performance and also the capability of a person to improve their performance with practice, using a VR marksmanship task. Performance enhancement was also investigated in [67], where the proposed control scheme provided virtually embodiable feedback during the control of a two-dimensional movement of a device. The results indicated that the absence of embodiable feedback led to higher control performance, greater discriminability in brain activity patterns, and increased cortical activation.

Four virtual classrooms were designed by the authors of [62], in order to investigate whether different window placement and room dimensions affect cognitive performance. In contrast with previous work no significant difference on working memory or mental fatigue was found. However, for the cognitive tasks involving short-term memory encoding, there were consistent and significant alterations in EEG features. Working memory performance was investigated in [64], using an experiment where a social (avatar) and non-social (stick) cue appeared in the VE and participants were asked to remember the location and the status of virtual objects. The performance on status information was found to be affected by both social and non-social cues, but the location information was altered only by the non-social cue.

Human communication was investigated in several studies, focusing on linguistic/verbal information and inter-brain synchrony/collaboration. Using an experiment where participants were immersed in a realistic 3D environment (virtual restaurant) a reliable N400 effect was observed when there was a mismatch between the visual and the auditory stimulus, showing that the combination of VR and EEG can be used to study language comprehension [69]. The effect of language on motor learning, where novel tools were presented with identifying labels, was investigated in [28]. Findings suggested that labels may strengthen one's memory and learning ability. A VR-based economic decision-making game was proposed by the authors of [55] to investigate if interpersonal touch and facial expressions influence participants perception and response to financial offers with different levels of fairness. The results suggest that at an early stage of processing fairness perception is not altered by the nonverbal context, but it can affect one's perception after the message is decoded.

Hyper-scanning is a neuroimaging technique, used to record neural activity from multiple participants at the same time, to investigate whether there is an interaction and synchronicity between their brains. Three pilot-studies were conducted in [72], where participants performed finger-pointing and finger-tracking exercises in pairs in two scenarios (real world and VR). Similar inter-brain synchrony was found to be induced by VR and the real-world.

Novel objects were presented to forty-three participants in study [32] and brain function was evaluated during the process of learning the function and usage of the objects. It was found that motor processing was influenced by object recognition. The results further support that object processing is not based only on previous experience, but is also shaped by the entire body interactions while solving a problem. In a similar direction, participants were asked to use familiar or abstract objects, both in real-world and AR, to create a scene [83]. The findings support that creativity cannot be evaluated for a single brain area but involves many regions of the brain. The influence of VR on creative performance was explored by the authors of [23]. The results revealed that VR can be a useful tool for the creative process, but it doesn't affect significantly the variety, novelty and creative outcomes.

The neural activity of different visual experiences during 2D and 3D video watching were measured in [80], where the proposed SVM classifier achieved a classification accuracy of 0.908.

### 4.5.2. Data Preprocessing and Artifact Removal

A notch filter was applied in a few papers [28,55,72] and FFT was used by [32]. A variety of band-pass filters was implemented as follows: 0.01–40 Hz [69], 0.1–40 Hz [31,32,44], 0.5–36 Hz [64], 0.5 Hz–50 Hz [27,30,62], 1–50 Hz [28], 0.5–60 Hz [72], 0.2–80 Hz [55], 1 Hz–80 Hz [83]. A combination of two band pass filters 1–100 Hz and 0.2–40 Hz [81] and a 40-Hz low-pass filter [55] were also applied. Other preprocessing methods included FFT [26], data augmentation [67], a surface Laplacian filter and TFRs, using a family of Morlet wavelets [32], an automated pre-processing pipeline proposed by Makoto and an approach proposed by Sareen et al. [72], PSD [83], frequency band-power and Partial Directed Coherence (PDC) and ROI analysis [62,83].

Artifacts were identified by visual inspection in several papers [27,30,55,64,81,83]. ICA was also an artifact rejection method preferred by a significant number of authors: [30,31,62,64,72,81,83]. While a combination of ICA and MARA was also used in study [55]. A number of authors removed specific channels or trials, regarded as artifactual such as trials during which the participants performed significant hand motions [23], data exceeding a 150 µV peak-to-peak amplitude criterion [44], a variety of voltage thresholds [28], components containing recordings from flat channels [42,44], signals that reflected EMG, ECG and EOG activity [44], channels with more than five SDs from the joint probability of the recorded electrodes [81], the signals from the frontal and prefrontal electrodes [32] and also those with less than 0.70 correlation with nearby channels [42].

Several other methods were implemented, namely the VMA and the MD [50], the Kurtosis method and equivalent dipole model fitting (DIPFIT) with a spherical four-shell (BESA) head model, the Brain Vision Analyzer software [69], the Autoreject algorithm and an individually tailored threshold-based artefact rejection procedure that used a stair climbing procedure [55], multiple source analysis and artifact subspace reconstruction (ASR) [44,62], the Irregular-Resampling Auto-Spectral Analysis method (IRASA) [44], ICLabel and ICA [44,62].

### 4.5.3. Signal Analysis

The mu rhythm, defined as the frequency band between 8 and 13 Hz, measured at central electrodes was investigated in studies [22,32]. All five frequencies δ, θ, α, β and γ were considered in a few studies [57,62,72]. Various subsets were also used α, β, and γ [80]; β and γ [14,32,71,82]; θ and α [64]; β [79]. In study [26] α and β2 frequency bands were investigated, the only case of sub-band consideration. Finally, different frequency ranges were considered rather than frequency bands by a significant number of authors [44,67,69,80].

### 4.5.4. Statistical Analysis

The Phase Locking Value (PLV) was calculated to measure inter-brain synchrony in [72], the HT and source localization, using eLORETA were also applied, while cross-spectral matrices were computed for each band of interest and for each subject. Average ERPs, the presence of a sustained N400 effect, repeated ANOVAs and the Greenhouse–Geisser correction were applied by the authors of [69]. Pairwise correlation comparison indicated four points and one ROI in [79]. LME models, fit by Restricted Maximum Likelihood (REML) estimates and type II Wald $\chi$2-tests were used in [44]. In study [64] Morlet wavelet transformation was applied to compute TFRs. ERPs and TFRs were considered in [26].

The VEPs and Movement-Related Cortical Potentials (MRCP) were analyzed in [81], using an automatic peak detection algorithm. The mean peak amplitudes were subjected to rmANOVA analysis. Tukey's HSD was employed for post hoc analysis, and in instances where sphericity was violated, corrected *p*-values were reported. Statistical analysis con-

sisted of the calculation of the Inter-Trial Coherence (ITC) and rmANOVA for the four ROIs in study [32]. Another study was focused on the analysis of the N1, (Medial Frontal Negativity) MFN, P3 and the Late Positive Potential (LPP), using rmANOVA [55]. The authors of [83], after verifying that equal variance could be assumed using Levene's test, they performed paired sample *t*-test, rmANOVA, pairwise PC. Corellograms were computed using Factor Analysis, PCA and the Bartlett's test. The amplitude of Prediction Error Negativity (PEN) and Pe were statistically analyzed with PCA and rmANOVA in [30]. Also, in article [27] the amplitude of PEN and P300 were evaluated with Multilevel Multivariate ANOVA (mmANOVA) and rmANCOVA.

### 4.5.5. Classification Methods

Feature extraction was performed in [67], using the Filter Bank Common Spatial Pattern (FBCSP) algorithm and Bayesian formulation of Fisher's LDA classification was applied on the features and a two-way ANOVA with 4-fold cross-validation were performed. The Mann–Whitney U test and BC were applied to investigate the ERD ratios. In study [28] ERD/ERS ratios were evaluated with pairwise comparison and two-tailed *t*-tests and a Monte Carlo cluster analysis was performed. A Convolutional Neural Network (CNN) based model (BCINet, [29]) was developed, followed by classification using 'softmax' and optimization with 'adam'. CC and pHRC datasets were divided into 60%, 20%, 20% for training, validation, and testing, respectively, for binary conditions using stratified sampling method. A neural network-based clustering approach was implemented in [31] to extract the PEN and Pe and the statistical analysis on the EEG data was conducted using ANOVA, ANCOVA, one-sample *t*-test, 1000-fold permutation test on PEN and Pe amplitude. A non-parametric KWT was performed by the authors of [62] for feature selection and a kernel SVM (k-SVM) was used as a classifier. A SVM classifier was also performed in [80], using an RF-based feature selection model.

### 4.6. Attention

Table 11 presents a summary of findings in the field of attention research.

### 4.6.1. Objectives and Outcomes

The effect of avatar ethnicity (white vs. black) and appearance (business man vs. beggar, with casual dress as a control condition) on the participants' alertness were studied in [34]. Higher levels of attention (alarm reaction) were measured when white participants were asked to help a white beggar or a black businessman and higher engagement when interacting with a black beggar or a white businessman. In a similar direction the authors of [70] measured increased alpha suppression, which is linked with greater attention allocation, when there was solidarity between the participant and the avatar in cases where the avatar was rated as average. An authoring tool (BARGAIN, [66]) for affective game design was proposed, which can be used to define rules for game adaptation according to the user's emotional state, resulting in an enhanced game experience. A study evaluated enhancements in both overall attention levels and the degree of engagement in young university students with a proposed 30-min software program (Virtual ART, [35]), inspired by Attention Restoration Theory (ART). An average score of approximately 85.37% accuracy of the proposed by [82] method, proved the feasibility of a real-time assessment of internal and external attention in an AR environment. The authors of [57] showed that goal directed attention may be assessed, using a saccadic eye experiment, where brain activity was evaluated through EEG analysis. The effects of restorative VEs on subject' attention and engagement for three creativity tests were assessed in study [75]. An increase in attention and engagement levels was associated with improvements on subjects' creativity.

**Table 11.** Attention.

| Authors, Year, Reference | Participants | Stimuli | Frequency Bands/ Range | Data Preprocessing | Artifact Removal | Classification Technique/Statistical Analysis | Main Findings |
|---|---|---|---|---|---|---|---|
| (Heyselaar et al., 2018) [70] | 30 | static photos, VR avatars | δ, θ, α, β | <150 Hz, TFR | VI, ICA | cluster randomization, ANOVA, Wald $\chi^2$ tests | • increased alpha suppression indicates greater attention allocation <br> • interaction with another individual influence attentional allocation |
| (Berger and Davelaar, 2018) [56] | 22 | Stroop task (VR vs. 2D) | α | FFT | α average power | Gratton effect, factorial ANOVA, Regression analysis | • large decreases in the Gratton effect reflect efficient training <br> • larger learning rates in the VR group compared to the 2D group. |
| (X. Yang et al., 2019) [45] | 60 | Virtual paintbrush with feedback | | | eSense algorithm | eSense algorithm | • participants who received reminder feedback had a higher attention value, significantly higher flow states, and higher quality creative product levels than those in the groups with no feedback or encouraging feedback |
| (Rupp et al., 2019) [33] | 10 | Attention and memory tasks (2D vs. VR) | | 0.1–50 Hz, 1–40 Hz, FFT | ICA, GA of ERP, ABM software | ERP | • no significant differences between 2D and VR |
| (Park et al., 2019) [57] | 15 | saccadic exercise | θ, α, β | 1–50 Hz | ICA, 80 μV threshold | PSD, ERSP, *t*-test, FCA | • goal directed attention may be assessed by EEG analysis |
| (Vortmann et al., 2019) [82] | 14 | VE (ring–sphere) | θ, α, β, γ | 1–50 Hz, notch, PSD | no artifact cleaning | hyperparameter optimization, LDA, Ledoit–Wolf lemma, ANOVA, 5-fold cross-validation | • it is feasible to perform real-time assessment of internal and external attention in AR environments using basic machine learning mechanisms. |

**Table 11.** *Cont.*

| Authors, Year, Reference | Participants | Stimuli | Frequency Bands/ Range | Data Preprocessing | Artifact Removal | Classification Technique/Statistical Analysis | Main Findings |
|---|---|---|---|---|---|---|---|
| (D'Errico et al., 2020)) [34] | 40 | VE | θ, α, β, β1, β2 | 3–40 Hz, PSD | ICA, SNR | (θ/β), (β2/β1), (β/(α + θ)) | • white participants exhibited greater attention towards white beggars or black businessmen and • greater engagement towards black beggars or white businessmen |
| (G. Li et al., 2020) [35] | 50 | Oddball task | θ, α | 0.5–30 Hz, notch | ICA, 100 μV threshold | P3b latency, ITC(θ), IEC(θ), ANOVA, paired *t*-test | • the proposed software improves attention and enhances engagement |
| (Benlamine et al., 2021) [66] | 29 | game | θ, α, β | | | Distraction = θ/β Engagement = β/(α + θ) | • emotion adaptive games enhance user experience |
| (Wan et al., 2021) [36] | 20 | game (2D vs. VR) N-back paradigm | Notch, EEMD | ICA, Wavelet threshold denoising | P300, LSTM | | • enhanced engagement in VR mode comparing with 3D mode • in VR mode males reached the peak of attention in less time than females |
| (Llinares et al., 2021) [37] | 160 | VE classroom | β, β2 | | | β, β2 relative power, Mann–Whitney | • cold-hued colors enhance working memory and attention |
| (Tian and Wang, 2021) [38] | 20 | Videos (2D vs. VR) | α, β | 0.1–95 Hz, notch, WT | ICA, 100 μV threshold | Mean energy, *t*-test | • participants attention in VR remains high even in repeated viewing, in contrast with 2D condition |
| (Cao et al., 2021) [46] | 32 | VE (interior, street, park) | α | 0.1–30 Hz | 20 μV threshold, ICA | ERP, Linear Mixed Effects Models, Type II Kenward–Roger test | • higher early visual attention for the VR condition • no differences on participants' recall functions |

**Table 11.** *Cont.*

| Authors, Year, Reference | Participants | Stimuli | Frequency Bands/ Range | Data Preprocessing | Artifact Removal | Classification Technique/Statistical Analysis | Main Findings |
|---|---|---|---|---|---|---|---|
| (Zhang et al., 2022) [73] | 16 | videos | θ, α | | | Granger causality, characteristic path length, GE, causal density and flow | • higher brain fatigue when using the HMD, rather than screen |
| (Chen et al., 2022) [87] | 28 | AR circles | θ, α, β | 0.1–30 Hz | VI, ICA, PSD, time series topography | α MI, α lateralization value, correlation analysis | • walking causes more distraction compared with standing tasks |
| (Darfler et al., 2022) [63] | 21 | visual memory task | θ, α, β | 0.5–50 Hz | ASR, ICA, ICLabel | ERSP, k-means, one-way ANOVA | • the presence of an avatar in a virtual classroom causes distraction and alters visual memory and performance |
| (H. Li et al., 2022) [75] | 72 | restorative VE | δ, θ, α, β, β1, β2, γ | 1–40 Hz, notch | VI, 150 μV threshold | PSD, rmANOVA, alertness = β2/β1, engagement = β/(θ + α) | • greater attention • creativity improvement |

ANOVA = Analysis Of Variance, ASR = Artifact Subspace Reconstruction, EEMD = Ensemble Empirical Mode Decomposition, ERD = Event-Related Desynchronization, ERP = Event-Related Potentials, ERSP = Event-Related Spectral Dynamics, GA = Grand Average, GE = Global Efficiency, ICA = Independent Component Analysis, IEC = Inter-trial Coherence, ITC = Inter-Trial Coherence, LDA = Linear Discriminant Analysis, LSTM = Long Short-Term Memory, MI = Modulation Index, PSD = Power Spectral Density, SNR = Signal to Noise Ratio, TFR = Time Frequency Representations, WT = Wavelet Transform.

A Stroop task was designed by the authors of [56], for the training of 22 participants to increase their level of α amplitude, which is correlated with attentional control. Feedback was provided to half of the participants in a 3D VE and the other half in a 2D environment. Larger learning rates, associated with larger decreases in the Gratton effect, were observed for the VE in comparison with the 2D environment. The feedback type was the objective of [45], where sixty participants received either no feedback, reminder feedback or encouraging feedback whenever EEG signal analysis showed inattention state, during a VR creativity task. Findings showed that participants who received reminder feedback had a higher attention value, significantly higher flow states, and higher quality creative product levels than those in the groups with no feedback or encouraging feedback. In a similar direction, comparative studies between 2D and VR were presented in papers [33,36] to assess the working memory state and attentional state of the participants. The results of [33] suggest that attention is not influenced by the extra equipment needed for the VE (HMD) and no significant differences in brain activity were found between the VR and 2D tasks. The findings of [36] indicate better performance and engagement in the VR mode. There was also a significant gender-related difference for the VE, where males reached the peak of attention in a shorter time than females. Higher brain fatigue was found to be induced in [73], while participants watched VR videos using HMD in comparison with a traditional 2D display.

In study [38], a repeated video viewing in 2D and VR were compared. Results suggested higher levels of attention in the VR condition which was at the same level for the repeated video viewing as well and stronger immersion. The authors of [46] designed 2D and VR tasks, where participants watched emotional or neutral films. Higher α activity was measured for the VR tasks, linked with increased early visual attention, while no differences on recall functions were found. A study focused on the effect of warm and cold hue-colored virtual classrooms on attention and memory functions [37], showed that cold hue colors improve performance in attention and memory tasks. A visual selective attention task was designed by the authors of [87], where participants were either standing or walking freely. The results showed that walking affects visual cortical processing.

Visual working memory, a core cognitive function associated with attention, was the objective of [63]. Findings suggest that visual memory in a VR classroom setting is affected by the presence of an avatar, which alters attentional focus.

### 4.6.2. Data Preprocessing and Artifact Removal

A notch-filter was applied at 50 Hz by [35,36,38,75,82]. A variety of band-pass filtering ranges of the data were applied; 3–40 Hz [34]; 0.5–30 Hz [35]; 0.5–10 Hz [63]; 1–40 Hz [33,75]; 1–50 Hz [33,57,82]; 0.1–30 Hz [46,87], 0.1–95 Hz [38], a 150 Hz low-pass filter was also used [70]. Several other preprocessing techniques were applied, including TFRs [70]; PSDs using Welch's method [34]; PSDs using the multitaper method, average and maximum power computation [82]; FFT [33,56]; WT [38] and the Ensemble Empirical Mode Decomposition (EEMD) [36].

Visual inspection was used by a small number of authors [70,75,87] to reject noisy channels. ICA was also implemented in many articles [33–35,38,46,63,70,87]. A small number of other techniques were also applied, including SNR [34], ASR and ICLabel [63], a voltage threshold of 150 μV [75], 100 μV [35,38], 80 μV [57] and 20 μV [46], the wavelet threshold denoising algorithm [36], a combination of the time-series signal, the signal topography, and the power spectrum [87], α average power [56], the NeuroSky Think Gear technology [45], ABM's proprietary software and the GA of ERP [33].

### 4.6.3. Signal Analysis

The authors of the following papers: [35,37,38,46,56,57,63,66,70,73,82,87] considered various subsets of the δ, θ, α, β and γ frequency bands, while in very few [34,75] the bands of interest also included β sub-bands. The signal was not decomposed into frequency bands by the authors of [33,35,36,45].

#### 4.6.4. Statistical Analysis

Several statistical analyses methods have been applied to assess participants' attention. The authors of [34] calculated three indexes: the calmness index ($\theta/\beta$), the alertness index ($\beta2/\beta1$) and the engagement index ($\beta/(\alpha + \theta)$). In a similar direction, the distraction index ($\theta/\beta$) which is negatively correlated with attention, was used in study [66]. A non-parametric cluster level randomization routine was used by the authors of [70] and resulted features were analyzed using an LME model. A step-wise "best-path" reduction procedure and the ANOVA function with Wald $\chi^2$ tests were also implemented.

Another research ([35]) focused on the computation of P3b latency, the frontal midline ITC and Inter-trial Coherence (IEC) in the theta frequency band. Subsequently, data were analyzed using ANOVA and paired $t$-tests.

Factorial ANOVAs and regression analysis on the Gratton difference were used in [56]. The participants' attention was measured via the NeuroSky brainwave device software (eSense) in study [45], while the products were evaluated by five researchers and the average value represented the creative quality of each product. According to another approach [37], four EEG metrics were calculated, based on the relative power of $\beta$ and $\beta2$ band from selected electrodes. The Mann–Whitney U test was applied. In study [38] the mean power of $\alpha$ and $\beta$ bands was compared for the 2D and VR tasks and for the repeated video viewing and $t$-tests were used for the comparison of subjective and objective data. Conclusions were drawn through ERP analysis in [33]. ERP analysis, a Linear Mixed Effects Model and Type II Kenward–Roger tests were applied in [46].

The construction of the EEG brain networks by Granger Causality and the computation of the characteristic path length, Global Efficiency (GE), causal density, and causal flow in the frequency domain of the brain effective network was selected from the authors of [73] for the statistical analysis of the signal. In a study presented in [87] the computation of $\alpha$ Modulation Index (MI) and $\alpha$ laterization value for the ERP and the time-frequency were considered. Subsequently, a cross-participant correlation analysis between ERP components, behavioral performance, and $\alpha$ MI was conducted. In study [57] ERSPs and PSDs were evaluated by $t$-tests and a Functional Connectivity Analysis was also implemented. The authors of [23] calculated the log values from the average changes in brain wave activity during each stage of the task and applied MANOVA and MANCOVA. In study [75] statistical analysis consisted of the calculation of PSD, rmANOVA and two additional measures: the alertness ($\beta2/\beta1$) and the engagement ($\beta/(\theta + \alpha)$) index. A k-means clustering and ERSP analysis along with one-way ANOVA was implemented in study [63].

#### 4.6.5. Classification Methods

A limited number of classification methods have been used for attentional assessment including LDA with 5-fold cross-validation, along with hyperparameter optimization, constrained classifier-Ledoit–Wolf lemma and ANOVA F-Value [82] and bidirectional Long Short-Term Memory (LSTM) neural network based on the BPTT algorithm and a 10-fold cross-validation, where attention features were extracted with the HHT method, split into two parts: empirical mode decomposition (EMD) and the HT [36].

### 5. Discussion

This systematic review includes research articles that employ EEG signal analysis to assess the cognitive state of participants who were immersed into VR, AR or MR environments, projected on HMDs. The research is conducted on the results of four established scientific databases, Scopus, ScienceDirect, IEEE Explore and PubMed. The first part of the review presents statistical results of the articles, such as publication year, number of participants, digital environment type, EEG equipment type, number of electrodes and objective area. In the second part, the papers are grouped into five main categories according to their field (cognitive load, immersion, spatial awareness, interaction with the digital environment and attention) and for each category the objectives and outcomes, data

preprocessing and artifact removal methods, EEG signal analysis and statistical analysis or classification methods are analyzed.

There are several important remarks derived from the review. Although the mean number of participants across all included articles was 33.8, the median is much lower being 25.5, since two studies [37,71] had a significantly larger number of participants (340 and 180 participants, respectively). Moreover, in 38% of the studies the total number of subjects is 20 or less, while in many cases the participants are divided into two groups, one in the physical environment and another one immersed in a virtual world, which leads in even narrower populations.

The vast majority of studies (~85%) use multi-electrode (>6) EEG equipment that record signals from multiple channels, offering the possibility of extracting more reliable measurements and more accurate conclusions on brain function. In addition, scientists have at their disposal a capable dataset to experiment on, and then extract the most useful characteristics for their research. However, fitting this type of EEG takes a lot of time and can cause extra fatigue to the participants, altering the research conclusions. This might also be the reason why the sample size is relatively small in the majority of the articles. Moreover, the increased number of electrodes can also increase the complexity of the calculations and add more artifacts to the recorded data, because of the HMD placement on top of the EEG electrodes. The relationship between the number of participants and the number of recording electrodes is depicted in Figure 7. The highest concentration of articles employs equipment with less than 20 electrodes and population with less than 50 participants. Furthermore, there is a significant number of articles that employ equipment with 57–64 electrodes and 16–50 participants. It is clear that a very small amount of research has been conducted for large numbers of participants (*y*-axis) or electrodes (*x*-axis).

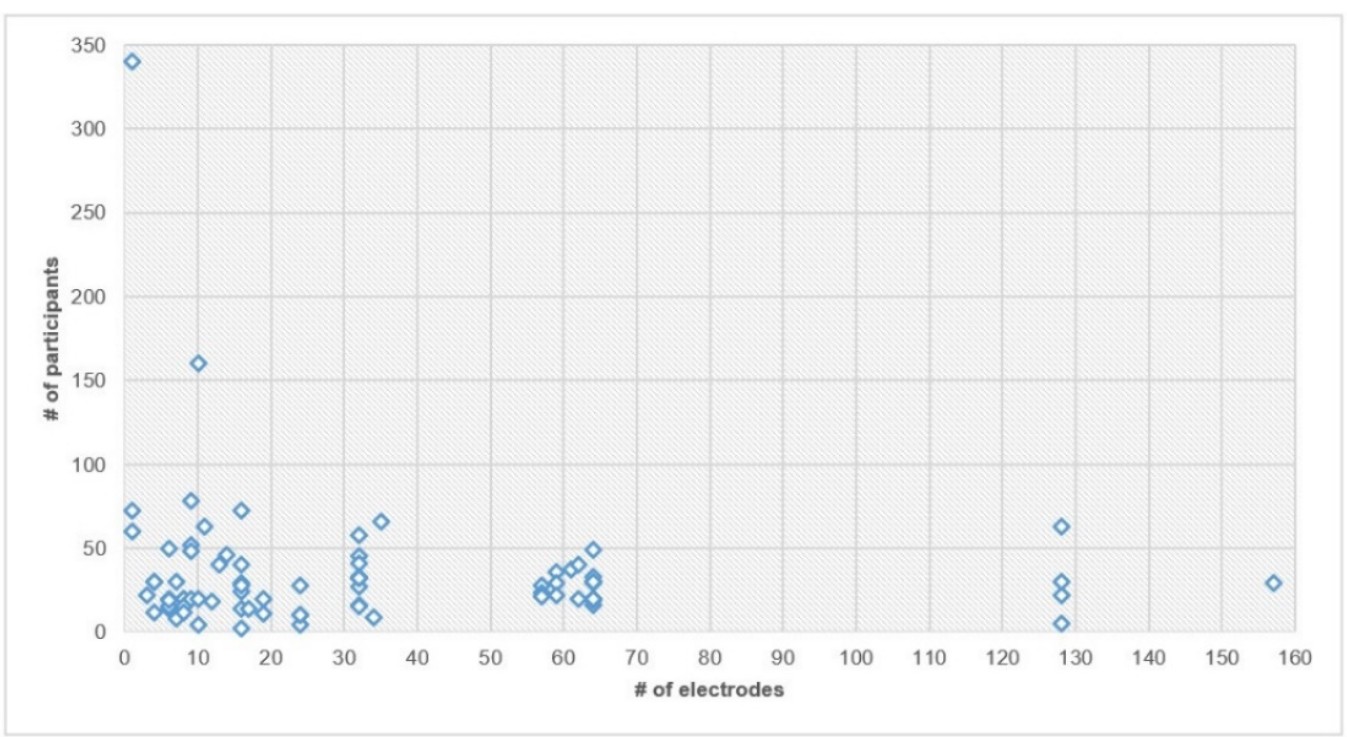

**Figure 7.** Number of participants vs. number of electrodes.

On the other hand, only 10% of all surveys investigated the cognitive effects of AR and there is no fully immersive MR experiment on cognition research. In a fully immersive MR environment the experiment setup does not include any 2D screens for the projection of the 3D elements. For example, the participants can wear an AR-HMD, while the VR

content can be viewed in a Computer-Aided Virtual Environment (CAVE), providing a much stronger sense of presence.

Regarding the frequencies that have been considered from the researchers, in a few cases [5,19,26,27,30,33,44,52,61,67,69,74,77,81] the signal frequency spectrum is not divided into frequency bands, while in most work the 5 basic bands (i.e., δ, θ, α, β, γ) or a subset of them are used [9–11,13–17,19,21–23,28,31,32,34,35,37–41,44,46,50,51,57,59,60,62,64,70–73, 75,79,80,83,85,88,89]. In recent years there has been a tendency to investigate α, β, and γ sub-bands, namely α1, α2, β1, β2, γ1, γ2 and γ3. A quite small number of articles [8,13,21, 26,34,37,39,47–49,75] have considered these sub-bands. Also, mu-rhythm, defined as the frequency band between 8 and 13 Hz and measured at central electrodes, was considered by the authors of [22,44]. Moreover, in study [13] the higher frequency range, termed as HF (70–100 Hz), was analyzed.

When it comes to artifact removal, visual inspection was a common method applied by a significant number of authors (~30%). Visual inspection could be useful for small datasets, but a sufficient automatic artifact rejection algorithm should rather be applied in large datasets. A limited number of methods for automatic artifact removal have been presented in the literature. Tremmel and Krusienski [11] reduced artifacts using a combination of two artifact suppression techniques: the Autoreject algorithm and an individually tailored threshold-based artefact rejection procedure that used a stair climbing procedure. Other automated artifact removal techniques were the BeMoBIL Preprocessing Pipeline [76], Artifact Subspace Reconstruction algorithm (ASR) [13,24,25,41,42,62,63,78] and EEGLAB automatic channel rejection tool [20], the algorithm developed by Gratton, Coles, and Donchin [71].

The techniques utilized to extract characteristics are presented in Table 12. These methods are categorized into Time-domain analysis, Frequency-domain Analysis, Time-frequency Analysis, Connectivity Analysis, Topographical Analysis and Nonlinear Analysis. It can be easily observed that there is a lack of research using Topographical Analysis and Nonlinear Analysis techniques, while the most popular seem to be the Time-domain analysis and Frequency-domain Analysis. These two categories also exhibit the highest diversity of distinct methodologies. Most studies employed statistical analysis methods for feature extraction, with fewer utilizing classification and clustering approaches (Table 13).

**Table 12.** Feature extraction methods.

| | | | |
|---|---|---|---|
| Time-domain Analysis | Amplitude Analysis | ERP, P300 amplitude, N1, P3, LPP, MFN, P3b, ERP mean amplitude, mean amplitudes (N1, P1, P3), N1, P2 mean and SD amplitudes, N400 | [19,26,27,33,52,55,69,74,77,84] |
| | Peak Analysis | VEP, MRCP, PEN, Pe, peak amplitudes (N1, P1, P3), first maximal negative deflection after T1 | [19,27,30,31,61,81] |
| | Area Under the Curve Analysis | Area under the average EEG graph | [85,86] |
| | Latency Analysis | P300 latencies, P3b amplitude latency | [35,52,84] |
| | Time-series Analysis | VAR | [14] |
| Frequency-domain Analysis | Spectral analysis | PSD, Welch | [13,15,16,40,57,75,76,83] |
| | Directed connectivity analysis | DTF, ffDTF | [14,53] |
| | Non-linear analysis | PLV | [72] |
| | Energy and power measures | Energy, sum of power, absolute band power, relative band power, power peak, power index | [13,17,37,38,47–49,59] |
| | Band power ratio measures | $\alpha/\beta$, $\theta/\alpha$, $\theta/\beta$, $(\theta+\alpha)/\beta$, $\theta/\beta$, $\beta2/\beta1$, $\beta/(\theta+\alpha)$ | [66,75,84] |
| | Specific band activity measures | $\theta$, $\alpha$, $\beta$ | [71,83] |
| | Other frequency domain measures | %$\theta$ change, ITC, IEC, $\alpha$ and $\beta$ wave difference, mean peak frequency | [20,32,35,53,65] |
| Time-frequency Analysis | | WPD, TFR, ERSP, ERD, ERS | [9,10,13,22,26,28,39,41,43,50,53,57,60,63,64,67,84] |
| Connectivity Analysis | | Functional Connectivity Analysis, Granger causality, characteristic path length, causal density and flow, GE, PC, PCA, SCoT, VAR | [14,22,30,57,73,79,83,87,88] |
| Topographical Analysis | | Scalp maps, IC Cluster analysis, ROI analysis | [43,52,59] |
| Nonlinear Analysis | | entropy | [41,84] |

**Table 13.** Feature analysis methods.

| | | | |
|---|---|---|---|
| Statistical analysis | Descriptive statistics | GA, sum of squares, mean of squares, Average log values, SD | [10,17,20,52,59,74,87] |
| | ANOVA methods | ANOVA, rmANOVA, rmANOVA, MANOVA, uANOVA, fANOVA, ANCOVA, MANCOVA | [15,19,20,22,23,27,30–32,35,43,47,49,52,53,55,56,59,61,69,75–77,81,83] |
| | Nonparametric statistics | two-tailed signed rank test, permutation test, Mann–Whitney U test, Cluster-based permutation testing, WSRT, KWT | [10,15,21,22,28,31,37,41,47–49,65,84] |
| | Parametric tests | *t*-test, Type II Wald x2-tests, regression analysis, pMFLR | [16,28,31,35,38,44,53,56–58,65,78,81,83–86] |
| | Post hoc tests | Post hoc Tukey HSD, post hoc Dunn's tests, Tukey's HSD, Newman–Kleus | [15,43,56,61,81] |
| | Other methods | Levene's test, Mauchly's test, pairwise comparisons, REML, Gratton effect, Greenhouse-Geisser correction, BC, sBEM, Tikhonov-regularized minimum-norm, F-tests type-III sum of squares, MI lateralization | [15,16,28,44,50,52,59,69,76,83,87] |
| Classification | Ensemble | CIT2FS, RF | [42,53] |
| | Artificial Neural Networks | CNN, LSTM | [28,31,36] |
| | Other Methods | SVM, LDA, LDFA, rLDA, KNN | [5,11,12,18,25,51,62,67,80,82,89] |
| Clustering | | k-means | [63] |

*Comparative Study*

Several efforts have been made in the literature to review works associated with cognitive assessment. However, only a few of these efforts have focused on assessment via electroencephalography and even fewer of them take immersive digital environments into account as a stimulus. Souza and Naves [91] presented a scoping review, focused on attention assessment in VEs. They reported results on the number of participants, the stimulus duration, EEG system variables (i.e., number of channels, sampling frequency, connectivity (wired/wireless), electrode locations), signal processing methods, data processing (user-dependent/independent, online/offline) and level of immersion. The articles were grouped into the following categories: attention allocation, workload, drive simulation, fatigue, game, VR variables, serious games.

The systematic meta-analysis of [92] evaluated human-environmental perception. The research investigated participant demographics (age, sex), the type of the recording signal (eye tracking, Electrodermal Activity (EDA)/EMG, EEG), the factors that influence environmental perception (color, sound, design, nature, point of vision, etc.) and the stimulus type (Real Environment, Virtual Screen, Semi Immersive, Virtual Immersive, V-Simulation, Images/Static, Real time Render). The authors of [93] provided an overview of various types of virtual reality (VR) used in biomedical practice and discussed their measurable effects on brain structure and cognitive performance. They also examined the important medical applications of VR technologies that enhance the quality of life for patients with neurological deficits. Finally, they discussed how the use of VR can benefit healthy individuals in terms of self-improvement and personal development. The cognitive abilities associated with human "narrative cognition" were explored in [94] both generally and specifically in relation to Mixed Reality Technologies.

Another research review [95] aimed to achieve several objectives, including clarifying the concept of design evaluation for human well-being, exploring available non-invasive methodologies for monitoring human neurological responses (such as fMRI, MEG, fNIRS, EEG), describing the specifications of the visual simulator used for virtual environments, and systematically reviewing existing literature that utilized empirical methodology for integrating an immersive visual virtual system with biometric data collection for architecture design evaluation pre- or post-occupancy.

This work investigated the cognitive effects of digital environments, projected on HMDs and assessed through EEG signal processing. To the extent of the authors' knowledge, there are no systematic reviews presented in the literature focused on this area. This review has a wide year range (2013–2022) and is focused on healthy subjects. In literature there are several reviews that investigate cognitive state of participants with pathological history or disorders, but very few are focused on general population and even less are systematic. As evidenced by Table 14, this systematic review covers the widest range of works and includes the largest number of articles, related to this area. Furthermore, this review can serve as a complete guide for researchers in neuroscience, as it is focused on the comprehensive presentation of experimental procedures, starting from the description of the experiment, the stages of signal processing, and finally, the key findings.

**Table 14.** Review articles on EEG cognitive assessment.

| Authors, Year, Reference | Review Type | Year Range | Articles Included | Main Objective | Sub-Categories | Conclusions |
|---|---|---|---|---|---|---|
| (Souza and Naves, 2021) [91] | Scoping | 2011–2020 | 40 | attention workload fatigue | • Attention allocation<br>• VR Variables<br>• Drive simulation<br>• Learning<br>• Workload<br>• Fatigue | • lack of standard terminology<br>• controlled trials may lead to biased conclusions<br>• HMD-VR can isolate external sources of distraction, so they could play a key role for research focused on internal attention<br>• Wide diversity in research methodologies and outcomes<br>• Gender bias<br>• through N100, N200, P100 and P300 in fronto-central-occipital brain areas and P300 in fronto-parietal brain areas are indicated for ERP analysis<br>• changes in $\beta/\theta$ ratio in fronto-parietal brain areas, $\theta$ in occipital and frontal area are the most informative for attention allocation research |
| (Shynu et al., 2021) [92] | Systematic | 2005–2020 | 44 | Environmental perception | • Influence of Real environment<br>• Impact of Virtual screen<br>• The effectiveness of a semi-immersive environment<br>• Virtual immersive based performance<br>• Consideration of age category<br>• Tools used for physiological readings<br>• the impact of viewpoint on the perception of the environment<br>• Impact of sound on an environment | • people prefer having elements of the natural environment in their living space<br>• differences in the perception of the environment related with gender, age, social, and other cognitive factors<br>• psycho-perception studies can be conducted using artificial environments<br>• more research is needed on age (especially elderly population) and gender influence in environmental perception<br>• the environmental design elements which could aid healthy and differentiable individuals has been very little investigated |

**Table 14.** *Cont.*

| Authors, Year, Reference | Review Type | Year Range | Articles Included | Main Objective | Sub-Categories | Conclusions |
|---|---|---|---|---|---|---|
| (Georgiev et al., 2021) [93] | Literature | 1984–2021 | 240 | Neurorehabilitation and Cognitive Enhancement | • Types of Virtual Reality<br>• Virtual Reality for Neurorehabilitation<br>• Virtual Reality for Replacement of Function<br>• Virtual Reality for Self-Enhancement | • the utilization of VR has potential advantages in areas such as training, research, and neurorehabilitation<br>• BCIs have the potential to aid in the restoration of lost functions, such as speech or mobility<br>• VR can enhance the sense of embodiment and enable precise control over bionic devices, which in turn can extend the capabilities of the human body |
| (Bruni et al., 2021) [94] | Literature | 1999–2021 | 98 | narrative cognition | • studies that directly or indirectly characterize aspects of narrative experience<br>• studies that investigate mixed reality experiences with or without narrative considerations | • further research should be conducted on the cognitive aspect of narrative engagement in mixed reality systems<br>• narrative cognition is considered a primary cognitive mode of humans for structuring experiences<br>• narratives in MRT and transmedia platforms can be used widely and in a variety of types |
| (Mostafavi, 2022) [95] | Systematic | 2015–2019 | 13 | spatial design evaluation | • architectural design evaluation environments<br>• human biometric feedback | • A complete protocol that documents the psychological comfort parameters of the experimental environment being monitored is necessary for advancing the field<br>• Insufficient knowledge of how humans respond to the physical environment can impede architects' ability to anticipate the various perceptions of a particular space resulting from their design choices |
| This study | Systematic | 2013–2022 | 63 | EEG cognitive assessment using HMD | • Cognitive load<br>• Immersion<br>• Spatial awareness<br>• Interaction with the digital environment<br>• Attention | • Further investigation is required in the fields of:<br>• immersive AR and MR applications<br>• automated artifact removal techniques<br>• signal frequency sub-bands<br>• classification methods<br>• efficiency of EEG headsets with small electrode number |

## 6. Conclusions

To the best of the authors' knowledge, there is currently no existing systematic review in the literature that specifically focuses on this area. This review encompasses a broad range of years (2013–2022) and primarily emphasizes research conducted on healthy subjects. This systematic review encompasses the widest array of studies and includes the largest number of articles within this field. Additionally, this review serves as a comprehensive guide for researchers in neuroscience, offering a detailed presentation of experimental procedures, including the number of participants, stimuli, frequency bands range, data preprocessing and EEG signal analysis.

From the study of the research papers included in this systematic review, various topics emerge that need to be further investigated. It was observed that the conclusions may be biased due to the small number of participants. Future research should expand the quantity of samples, to avoid bias in experimental results and to increase reproducibility. Special attention should be paid to the fact that people's cognitive abilities are affected by age, so elderly people may not be the best fit for general-purpose experiments i.e., experiments that do not study cognitive abilities in relation to age.

The authors of [21] suggested that any VR application, needs to last at least for 42.8 s in order to be effective, meaning that the exposure time in the MR environment should be sufficient in order to achieve the participants' illusion of being immersed into a close-to-real environment. On the other hand, due to VR motion sickness caused by HMDs, an upper bound should be set for the duration of the experiments, for the limitation of misleading results. Another direction in this field could be the optimization of motion related algorithms to limit the effect of the unpleasant feeling of illness.

A diverse range of HMD types is presented in Table 3. It is crucial to recognize that the choice of HMD can exert a substantial impact on cognitive assessment outcomes, influencing critical elements such as visual immersion, user comfort, interactivity, sensory input, and the overall user experience. For researchers engaged in cognitive assessments within virtual environments, a meticulous evaluation of these factors during HMD selection is essential to guarantee the validity and reliability of their assessments.

Regarding the EEG signal analysis methods, there is a gap in the research of establishing a sufficient automated artifact rejection pipeline that would not need any human interference, such as visual inspection. Also, very few researchers have focused on $\alpha$, $\beta$ and $\gamma$ sub-bands. Continued exploration of $\alpha$, $\beta$ and $\gamma$ sub-bands may also contribute to advancing our understanding of cognitive assessment, shedding light on aspects such as attention, memory, perception, and executive functions.

Statistical analysis on EEG recordings is a robust method for assessment, because it can provide insights for the understanding of the relationship between EEG variables, it can handle missing values or noisy data and can be an excellent tool to test hypotheses. However, classifiers can provide more qualitative results for EEG data with complex patterns, they can provide automated results and can be used to develop BCI applications. They can also serve as a research prediction tool and handle large datasets more efficiently. Only a small percentage (~20%) of the studies utilized classification methods to draw their conclusions. Further investigation into the use of these methods could facilitate research in the area of this review.

Despite the extensive literature available on the use of VR in an experimental setting, research focused on AR and MR environments remains notably limited. Future research in this direction could fill the gap between the new immersive technologies and the current knowledge of their effects on human cognition. On the other hand, in several research papers, MR environments are designed using HMD equipment for the AR part of the application and a 2D monitor for the VR application. They are also called monitor-based (non-immersive) video displays [96]. A highly immersive MR environment is considered to include an immersive 3D virtual environment that can be produced in specially designed rooms (e.g., CAVE system) where the VE is projected on three or four walls. Whether the combination of VR and AR truly adds value in human cognition remains an open question.

There is also a lack of publicly available EEG datasets in digital environments, created for the study of human cognition. More data availability can enhance the investigation of brain function in this field.

A limited number of articles support that reliable conclusions can be drawn using headband or cap-type EEG equipment, which use a smaller number of electrodes and involve a much easier installation process on the scalp. Experiments using both EEG types for the same stimuli and setup can provide the research community with more evidence on the lowest number of electrodes sufficient to draw reliable results.

According to [97], instructional methods should avoid overloading working memory with extraneous activities that do not directly contribute to learning, as working memory has limited capacity. Several studies considered an increase in cognitive load as overloading or as cognitive burden, though others have linked this increase with higher levels of immersion or attention. More research has to be conducted to determine how the outcomes reflect positive or negative effects, as well as the circumstances under which the utilization of digital environments can be regarded as cognitively advantageous rather than being a cause of fatigue and distraction.

**Author Contributions:** Conceptualization, F.G., K.D.T., P.A. and M.G.T.; methodology, F.G., N.G. and M.G.T.; validation, M.G.T.; data curation, F.G.; writing—original draft preparation, F.G.; writing—review and editing, F.G., K.D.T., P.A., N.G. and M.G.T.; visualization, F.G.; supervision, M.G.T.; project administration, M.G.T.; funding acquisition, M.G.T. All authors have read and agreed to the published version of the manuscript.

**Funding:** This research was funded in part by the project "AGROTOUR–New Technologies and Innovative Approaches to Agri-Food and Tourism to Boost Regional Excellence in Western Macedonia" (MIS 5047196), which is implemented under the Action "Reinforcement of the Research and Innovation Infrastructure", supported by the "Operational Program Competitiveness, Entrepreneurship and Innovation" (NSRF 2014–2020), and co-financed by Greece and the European Union (European Regional Development Fund).

**Data Availability Statement:** Data sharing not applicable.

**Conflicts of Interest:** The authors declare no conflict of interest. The funders had no role in the design of the study; in the collection, analyses, or interpretation of data; in the writing of the manuscript; or in the decision to publish the results.

## Abbreviations

| | |
|---|---|
| AMICA | Adaptive Mixture ICA |
| ANOVA | Analysis of Variance |
| AR | Augmented Reality |
| ASR | Artifact Subspace Reconstruction |
| BC | Bonferroni Correction |
| BCI | Brain Computer Interface |
| CAR | Common Average Referencing |
| CNN | Convolutional Neural Network |
| CSP | Common Spatial Pattern |
| DFT | Discrete Fourier Transform |
| ECG | Electrocardiogram |
| EDA | Electrodermal Activity |
| EEG | Electroencephalogram |
| EEMD | Ensemble Empirical Mode Decomposition |
| EKG | Electrocardiogram |
| eLORETA | exact Low-Resolution Electromagnetic Tomography |

| | |
|---|---|
| EMG | Electromyogram |
| EOG | Electrooculogram |
| ERSP | Event-Related Spectral Perturbation |
| ERP | Event-Related Potential |
| FBCSP | Filter Bank Common Spatial Pattern |
| FFT | Fast Fourier Transform |
| ffDTF | full frequency Directed Transfer Function |
| fMRI | functional Magnetic Resonance Imaging |
| GA | Grand Average |
| GE | Global Efficiency |
| GSR | Galvanic Skin Response |
| HMD | Head Mounted Display |
| HR | Heart Rate |
| HSD | Honest Significant Difference |
| HT | Hilbert Transform |
| ICA | Independent Component Analysis |
| IEC | Inter-trial Coherence |
| IRASA | Irregular-Resampling Auto-Spectral Analysis |
| IS | Independent Samples |
| ITC | Inter-Trial Coherence |
| KWT | Kruskal–Wallis test |
| LDA | Linear Discriminant Analysis |
| LDFA | Linear Discriminant Function Analysis |
| LME | Linear Mixed Effects |
| LPP | Late Positive Potential |
| LSTM | Long Short-Term Memory |
| MAD | Mean Absolute Distance |
| MANOVA | Multivariate Analysis Of Variance |
| MANCOVA | Multivariate Analysis of Covariance |
| MARA | Multiple Artifacts Rejection Algorithm |
| MD | Mahalanobis Distance |
| MI | Modulation Index |
| MFN | Medial Frontal Negativity |
| MMANOVA | Multilevel Multivariate ANalysis Of VAriance |
| MR | Mixed Reality |
| MRCP | Movement-Related Cortical Potentials |
| PCA | Principal Component Analysis |
| PC | Pearson Correlation |
| PEN | Prediction Error Negativity |
| PLV | Phase Locking Value |
| pMFLR | penalized Multiple Functional Logistic Regression |
| REML | REstricted Maximum Likelihood |
| RF | Random Forest |
| rmANOVA | repeated measures ANOVA |
| ROC | Receiver Operating Characteristics |
| ROI | Region of Interest |
| sBEM | symmetric Boundary Element Method |
| SD | Standard Deviation |
| SNR | Signal-to-Noise Ratio |
| SVM | Support Vector Machine |
| TFR | Time-Frequency Analysis |
| VI | Visual Inspection |
| VE | Virtual Environment |
| VEP | Visual Evoked Potentials |
| VMA | Variance of the Maximal Activity |
| VR | Virtual Reality |
| WSRT | Wilcoxon Signed-Rank Test |
| WT | Wavelet Transformation |

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
