# Peer review of "Cognitive Assessment Based on Electroencephalography Analysis in Virtual and Augmented Reality Environments, Using Head Mounted Displays: A Systematic Review"

_2504-2289, doi:10.3390/bdcc7040163_

Round 1

Reviewer 1 Report

The manuscript entitled: Cognitive Assessment Based on Electroencephalography Anal-2 ysis in Virtual and Augmented Reality Environments, Using 3 Head Mounted Displays: A Systematic Review, is a very interesting systematic review of the current field of VR/AR/MR devices used to stimulate cognitive functions and assessed through electroencephalography (EEG), which fits nicely into the journal’s objectives.

I commend the work of the authors and suggest a number of things to consider to improve the manuscript before acceptance.

Abstract

In the second paragraph the authors wrote “Notably, there has been a scarcity of research examining the effects of AR and MR on human cognition.” This does not seem to be the case since you have 82 articles in your review, not considering the fact that you excluded other reviews and books. I would remove or reword that sentence.

The last paragraph is almost irrelevant. I would instead add important results within the 5 cognitive areas you looked into: attention, cognitive load, immersion etc. and expand if possible on the limitation of the EEG as a method for cognitive assessment.

1.     Introduction

I believe the Introduction section can be improved if you start the section with the topic of your review, that is, cognitive assessment. Instead, the authors have chosen to start with the solution statement, that is, using VR/AR/MR to study cognitive assessment. The problem statement, i.e., difficulty of doing cognitive assessment in real life experiments comes also late. Please, follow a clearer structure like: what is the paper about? What is the problem? How can you solve it? Why bother? What is novel? A restructure would make the flow easier to follow.

Page 2 line 68. You need a reference.

Page 2 line 70-71. You have the same acronym EEG for two separate terms.

Page 3 line 84-86. You name sub-bands without explanation for what they entail.

I believe you followed a research question. Please state it.

2.     Research methodology

I commend the authors for uploading the protocol on OSF. Not many do that!

You may want to use Inclusion criteria and Exclusion criteria. Now the go into each other and difficult to follow.

3.     Study statistics.

Figure 4. You have number of studies in both axes.

Figure 7. same here

Figure 8. Missing legend on the X axis, i.e., # of studies

4.     Results

This is a good section with details results of the 82 included articles divided into the 5 fields of cognitive functions, and for each, the outcomes further divided into data pre-processing and artifact removal methods, the frequency bands and the classification or statistical analysis. Good work!

Up to page 11 the sequence of the pagination is correct. Then it gets messy and difficult to point out exactly to other amends.

5.     Discussion

Second page of the Discussion. Towards the end of the page you mentioned again the sub-bands, but you do not give an explanation to what it entails, simply that they have looked at them. Please provide discussion points.

On the third page of the Discussion and before Table 12, you present new results about extraction analysis. Those are not presented in the Results section. Why? Should they not be presented in the Results first?

Section 5.1 line 12. There is a round apostrophe too much before [92].

6.     Conclusions

You take up the issue of VR motion sickness. This is also an important issue to talk about up front in the Introduction, as this is probably one of the major cause to why VR is not wide spread or not suitable for experimental studies.

First page line 76. Very few research on the effect of VR and MR is vague. Effects of what? Robert Stone has written extensively about these and is very critical. You have to be specific.

Future Trends section is a repetition of the Conclusions. Can be removed.

I would move the index at the beginning as there are acronyms that you use in the paper which are not spelled out. BCI is one example.

An extra proofreading will make the flow a little better and catch some of the small language and syntax misstakes throughout 

Author Response

We appreciate the time and effort the reviewer have dedicated to reviewing our paper. We value the insights provided and have carefully considered each of their points. All the revised parts of the manuscript are highlighted in grey color.

Comment:

(abstract) In the second paragraph the authors wrote “Notably, there has been a scarcity of research examining the effects of AR and MR on human cognition.” This does not seem to be the case since you have 82 articles in your review, not considering the fact that you excluded other reviews and books. I would remove or reword that sentence.

Answer:

The sentence has been removed from the revised version of the manuscript.

Comment:

(abstract) The last paragraph is almost irrelevant. I would instead add important results within the 5 cognitive areas you looked into: attention, cognitive load, immersion etc. and expand, if possible, on the limitation of the EEG as a method for cognitive assessment.

Answer:

We have removed the last paragraph and replaced it according to this comment. (page 1, lines 22-30)

Comment:

I believe the Introduction section can be improved if you start the section with the topic of your review, that is, cognitive assessment. Instead, the authors have chosen to start with the solution statement, that is, using VR/AR/MR to study cognitive assessment. The problem statement, i.e., difficulty of doing cognitive assessment in real life experiments comes also late. Please, follow a clearer structure like: what is the paper about? What is the problem? How can you solve it? Why bother? What is novel? A restructure would make the flow easier to follow.            

Answer:

Based on your suggestion, we have restructured the introduction in order to be more comprehensive for the reader (pages 1-3, lines 35-106). Furthermore, information has been added for the EEG frequency sub-bands, according to another comment.

Comment:

Page 2 line 68. You need a reference.

Answer:

Since the introduction was fully reconstructed, the statement in line 68 has been removed.

Comment:

Page 2 line 70-71. You have the same acronym EEG for two separate terms.

Answer:

The acronym EEG was removed from the term “electroencephalography” (page 1, line 44)

Comment:

Page 3 line 84-86. You name sub-bands without explanation for what they entail.

Answer:

The following brief explanation of each sub-band has been added for better context (page 2, lines 56 – 65):

Comment:

I believe you followed a research question. Please state it

Answer:

This work is a systematic review on cognitive assessment based on EEG analysis digital environments, projected on Head Mounted Displays, for healthy individuals and the research mainly focused on identifying and analyzing all related literature (as described in the research methodology in Fig. 2, page 4) and to identify research gaps and future trends.

Comment:

I commend the authors for uploading the protocol on OSF. Not many do that!

Answer:

Thank you for your kind comment.

Comment:

You may want to use Inclusion criteria and Exclusion criteria. Now the go into each other and difficult to follow.

Answer:

We have restructured the manuscript in order to clearly present the inclusion and exclusion criteria. (page 4, lines 137 – 160)

Comment:

  • Figure 4. You have number of studies in both axes.
  • Figure 7. same here
  • Figure 8. Missing legend on the X axis, i.e., # of studies

Answer:

We have made all necessary revisions to address the issues you pointed out. Figures 4, 7 and 8 had lost significant information because their style had been changed. The figures 4 (page 5), 7 (page 9), and 8 (page 10) have been restored to their original state, providing all the missing information.

Comment:

(results) This is a good section with details results of the 82 included articles divided into the 5 fields of cognitive functions, and for each, the outcomes further divided into data pre-processing and artifact removal methods, the frequency bands and the classification or statistical analysis. Good work!

Up to page 11 the sequence of the pagination is correct. Then it gets messy and difficult to point out exactly to other amends.

Answer:

We sincerely appreciate your positive feedback and we are pleased to hear that you found the section well-structured. The pagination after page 11 is now corrected and We apologize for any inconvenience this may have caused.

Comment:

Second page of the Discussion. Towards the end of the page you mentioned again the sub-bands, but you do not give an explanation to what it entails, simply that they have looked at them.

Answer:

In the introduction section, we have added a comprehensive overview of the sub-bands, their frequency ranges, and their significance in the context of our research objectives (page 2, lines 55-65). Also, the following text: “Continued exploration of α, β and γ sub-bands may also contribute to advancing our understanding of cognitive assessment, shedding light on aspects such as attention, memory, perception, and executive functions.” Has been added to the conclusions section (page 48, 5th paragraph)

Comment:

On the third page of the Discussion and before Table 12, you present new results about extraction analysis. Those are not presented in the Results section. Why? Should they not be presented in the Results first?

Answer:

On the third page of the Discussion (page 42), we have categorized the methods used by all researchers (also mentioned in the results section) regarding signal processing characteristics. As a result, the following method categories emerged: Time-domain analysis, Frequency-domain Analysis, Time-frequency Analysis, Connectivity Analysis, Topographical Analysis, and Nonlinear Analysis.

Comment:

Section 5.1 line 12. There is a round apostrophe too much before [92].

Answer:

The redundant parenthesis has been deleted.

Comment:

(Conclusions) You take up the issue of VR motion sickness. This is also an important issue to talk about up front in the Introduction, as this is probably one of the major cause to why VR is not wide spread or not suitable for experimental studies.

Answer:

The paragraph in page 2 (lines 86 – 90) is extended with the following text “However, a very serious drawback in conducting experiments through VR is the sensation of dizziness and nausea, known as VR sickness, often experienced by participants. This phenomenon can significantly affect cognitive state and, consequently, the experiment's results. This is one of the key reasons why research involving VR is not wide spread and has a restricted number of participants.”

Comment:

(Conclusions) First page line 76. Very few research on the effect of VR and MR is vague. Effects of what? Robert Stone has written extensively about these and is very critical. You have to be specific.

Answer:

We acknowledge that our previous statement may have been vague, and we rephrased this sentence as follows: “Despite the extensive literature available on the use of VR as an experimental setting, research focused on AR and MR environments remains notably limited. Future research in this direction could fill the gap between the new immersive technologies and the current knowledge of their effects on human cognition.” (page 48, lines 91-94)

Comment:

Future Trends section is a repetition of the Conclusions. Can be removed.

Answer:

The Discussion section was accidentally replicated. This has been corrected in the revised version of the manuscript.

Comment:

I would move the index at the beginning as there are acronyms that you use in the paper which are not spelled out. BCI is one example.

Answer:

Some acronyms were not addressed for the first time they were mentioned. All acronyms were checked and the manuscript was corrected.

Comment:

An extra proofreading will make the flow a little better and catch some of the small language and syntax mistakes throughout 

Answer:

The manuscript has undergone additional proofreading, and minor mistakes have been corrected.

Reviewer 2 Report

The comments are,

1.     Highlight the objective of this study. It has given in paragraph rather the specific highlighted points.

2.     How the signal analysis carried? Is it used any tool or instruments for detecting signal variation?

3.     Suggested to draw the common framework/flow chart to replace Figure 2. PRISMA flow chart. Authors used quantitative values inside the flow chart.

4.     Mention the axis unit values of Figure 8. Objective area.

5.     This 4.3.1. Objectives and outcomes are not matching with abstract.

6.     Few repeated terms like Signal analysis; Statistical analysis should be more specific. Author should make this terms in domain specific. Example,. Signal analysis in…

Author Response

We appreciate the time and effort the reviewer have dedicated to reviewing our paper. We value the insights provided and have carefully considered each of their points. All the revised parts of the manuscript are highlighted in grey color.

Comment:

Highlight the objective of this study. It has given in paragraph rather the specific highlighted points.

Answer:

Thank you for your constructive feedback. The introduction has been reconstructed and the objective is highlighted in the last paragraph (page 3, lines 99-106)

Comment:

How the signal analysis carried? Is it used any tool or instruments for detecting signal variation?

Answer:

The objective of this study is to perform a systematic review of existing research on cognitive assessment, based on EEG analysis in digital environments, using HMDs. All signal analysis that has been presented in the literature are analyzed in sections 4.x.3

Comment:

Suggested to draw the common framework/flow chart to replace Figure 2. PRISMA flow chart. Authors used quantitative values inside the flow chart.

Answer:

We have opted to utilize the PRISMA flow chart for our study as it aligns with the systematic review nature of our research. This decision is consistent with the official PRISMA guidelines, which mandate the inclusion of quantitative values within the PRISMA flow diagram, as detailed in the PRISMA Statement (http://www.prisma-statement.org/Default.aspx) and the associated Flow Diagram (http://www.prisma-statement.org/PRISMAStatement/FlowDiagram).

 Comment:

Mention the axis unit values of Figure 8. Objective area.

Answer:

We appreciate the reviewer for bringing this to our attention. The values have been incorporated into the revised version of Figure 8.

Comment:

This 4.3.1. Objectives and outcomes are not matching with abstract

Answer:

We have revised the abstract so as be in line with all cognitive functions that are evaluated in this systematic review (page 2, lines 92-96)

Comment:

Few repeated terms like Signal analysis; Statistical analysis should be more specific. Author should make this terms in domain specific. Example,. Signal analysis in…

Answer:

In all instances, the signal refers to EEG data. We have made multiple revisions to the text to enhance clarity in this regard.

Reviewer 3 Report

Conclusion and future trends are almost the same, need to change

Author Response

We appreciate the time and effort the reviewer have dedicated to reviewing our paper. All the revised parts of the manuscript are highlighted in grey color.

Comment:

Conclusion and future trends are almost the same, need to change

Answer:

The Discussion section was accidentally replicated. This has been corrected in the revised version of the manuscript.

Reviewer 4 Report

This article is a review paper for HMD and EEG-based cognitive assessment for VR/AR/MR environments.

The article describes a wide range of review which are helpful for other researchers on the focused areas.

To improve the article, I suggest the following:

1. There are various types of HMD (see-through style, mobile style, camera-attached AR style, glass style, eye tracking capability, and different resolutions).

I think the types of HMD affect the human visual cognitive process, especially in VR motion sickness. 

However, the review does not consider the HMD types. 

To improve the article it would be better to comment the device related effects on congnitive assessment.

2. The article says that the review protocol has been registered with the OSF

at https://osf.io/svrxh/?view_only=11f344e6174d4dcf8ea2562bbdf68e12. When I checked the URL, the OSF Storage cotains none. It should be checked again to ensure there are valid files.

Also, it would be better to replace the reference [6] texts with the corresponding URL.

Author Response

We appreciate the time and effort the reviewer have dedicated to reviewing our paper. We value the insights provided and have carefully considered each of their points. All the revised parts of the manuscript are highlighted in grey color.

Comment:

There are various types of HMD (see-through style, mobile style, camera-attached AR style, glass style, eye tracking capability, and different resolutions). I think the types of HMD affect the human visual cognitive process, especially in VR motion sickness. However, the review does not consider the HMD types. To improve the article it would be better to comment the device related effects on congnitive assessment.

Answer:

The reviewer has raised a very interesting aspect. In the papers that were analyzed during this work there is limited effort on evaluating the effect of using different HMD devices for the same cognitive experiment. We have added a comment in section 3.3 (page 6, lines 182-186) the discussion section, clearly identifying this as an open research issue (page 48, lines 70-75).

Comment:

The article says that the review protocol has been registered with the OSF at https://osf.io/svrxh/?view_only=11f344e6174d4dcf8ea2562bbdf68e12. When I checked the URL, the OSF Storage contains none. It should be checked again to ensure there are valid files.

Answer:

The correct OSF link (https://osf.io/kfx5p), incorporating the research protocol, has been inserted into the manuscript (page 3, line 114).

Comment:

Also, it would be better to replace the reference [6] texts with the corresponding URL

Answer:

The reference has been formatted in accordance with the guidelines outlined in OSF's instructions, as detailed in: https://help.osf.io/article/221-generate-citations.

Reviewer 5 Report

1.   A short summary of the manuscript explaining what the study is about:

The study proposes valuable research papers that utilize EEG signal analysis to evaluate the cognitive condition of individuals who have been immersed in virtual reality (VR), augmented reality (AR) or mixed reality (MR) environments displayed on head-mounted displays (HMDs). The investigation draws from data found in four well-established scientific databases: Scopus, ScienceDirect, IEEE Explore, and PubMed. The initial section of the review presents statistical findings from the articles, including publication year, participant count, type of digital environment, EEG equipment type, number of electrodes, and the specific cognitive domain under examination. In the subsequent part, the papers are categorized into five primary areas based on their focus (cognitive workload, immersion level, spatial awareness, interaction with the digital environment, and attention). For each category, the study objectives, outcomes, data preprocessing techniques, artifact removal methods, signal analysis approaches, and statistical or classification methodologies are thoroughly examined.

It is advantageous that: Most of the reference in the literature review are from the last 5 years and the review protocol has been registered with the Open Science Framework (url: 101 https://osf.io/svrxh/?view_only=11f344e6174d4dcf8ea2562bbdf68e12).

However, the findings are not clearly presented from neuroscience perspective, and the readability of the study on cognitive assessment based on electroencephalography analysis in virtual and augmented reality environments, utilizing three head mounted displays, is not optimal.

2.     Issues found that need to be addressed:

2.1. Major issues:

·       Future trends (rows 103 to 155) are the same as those in Conclusions (rows 48 to 100)?

·       In the abstract authors claim in line 27: offering a detailed presentation of experimental procedures, including the description of the experiment. The tables don’t contain info about the experiment. It should be replaced by - including the number of participants, stimuli, frequency bands range, data preprocessing.

·       Although Table 3. considers digital environment type, Table 4. is about EEG type, Table 5.  is about distribution of number of electrodes utilized by researchers, there is still lack of information about EEG equipment and VR/AR/MR device used. Replication of the findings from experiments could be facilitated if these device exist in the second part of the paper. In all tables, grouped into five main categories, two new categories can be added for EEG device and VR/AR/MR device used.

·       A lot of the Main Findings in the tables are not well structured and don’t present the neuroanalysis. For instance,  [29] in Table 8:

The authors’ main finding is: immersion assessment experiments may be designed using irrelevant stimuli as a distractor – nonsense?

It should be something like: Immersion was characterized as the focused attention on external auditory stimuli unrelated to the game, and it was assessed indirectly by analyzing ERPs in response to an auditory oddball task.

·       The sentence at the end of Section 4.5.1. “The neural activity of different visual experiences during 2D and 3D video watching were measured in [60], where the proposed SVM classifier achieved a classification accuracy of 0.908.” is also not a result?

·       In Section 4.6.1. References [64] is not analyzed, while the main findings about neuroanalysis in  [65] and [66] should be reconsidered.

·       In Section 5. Authors claim: the papers are grouped into five main categories ac-cording to their field (cognitive load, immersion, spatial awareness, interac-tion with the digital environment and attention) and for each category the objectives and outcomes, data preprocessing and artifact removal methods, signal analysis and statistical analysis or classification methods are analyzed.

From my point of view, these data in subsections (x.x.3 to x.x.5) are just mentioned and grouped. Since this information exist in the tables, the subsections for the preprocessing and artifact removal methods, signal analysis and statistical analysis or classification methods, have to be removed. Real analysis of data is presented in Figures and in the tables in Section 5 (Tables 12, 13 ,14).

 2.2.Minor issues:

Some of references have DOI, some not.

Minor editing of English language required

Author Response

We appreciate the time and effort the reviewers have dedicated to reviewing our paper. We value the insights provided and have carefully considered each of their points. All the revised parts of the manuscript are highlighted in grey color.

Comment:

Future trends (rows 103 to 155) are the same as those in Conclusions (rows 48 to 100)?

Answer:

The Discussion section was accidentally replicated. This has been corrected in the revised version of the manuscript.

Comment:

In the abstract authors claim in line 27: offering a detailed presentation of experimental procedures, including the description of the experiment. The tables don’t contain info about the experiment. It should be replaced by - including the number of participants, stimuli, frequency bands range, data preprocessing.

Answer:

The abstract has been updated according to the reviewer's comment (page 1, lines 17-19).

Comment:

Although Table 3. considers digital environment type, Table 4. is about EEG type, Table 5.  is about distribution of number of electrodes utilized by researchers, there is still lack of information about EEG equipment and VR/AR/MR device used. Replication of the findings from experiments could be facilitated if these device exist in the second part of the paper. In all tables, grouped into five main categories, two new categories can be added for EEG device and VR/AR/MR device used.

Answer:

We would like to thank the reviewer for this valuable comment. We acknowledge that the EEG and HMD device information is important, therefore we have included related information for each paper (page 6, table 3; page 8, table 4) providing a clear oversight of the used devices to the potential reader.

Comment:

A lot of the Main Findings in the tables are not well structured and don’t present the neuroanalysis. For instance, [29] in Table 8:

The authors’ main finding is: immersion assessment experiments may be designed using irrelevant stimuli as a distractor – nonsense?

It should be something like: Immersion was characterized as the focused attention on external auditory stimuli unrelated to the game, and it was assessed indirectly by analyzing ERPs in response to an auditory oddball task.

Answer:

The manuscript has been updated according to the reviewer’s comment. The proposed sentence “Immersion was characterized as the focused attention on external auditory stimuli unrelated to the game, and it was assessed indirectly by analyzing ERPs in response to an auditory oddball task” has replaced the sentence “immersion assessment experiments may be designed using irrelevant stimuli as a distractor”  (Table 8, page 21).

Comment:

The sentence at the end of Section 4.5.1. “The neural activity of different visual experiences during 2D and 3D video watching were measured in [60], where the proposed SVM classifier achieved a classification accuracy of 0.908.” is also not a result?

Answer:

We have included this information in the results. The respective information has been added in the results column of table 10 (page 31).

Comment:

In Section 4.6.1. References [64] is not analyzed, while the main findings about neuroanalysis in [65] and [66] should be reconsidered.

Answer:

The analysis for reference [64] is added in the first paragraph of section 4.6.1 (page 34) in the revised version of the manuscript. Also, [65] and [66] has been revised according to the reviewer’s comment in page 34 and also in the respective table 11 (page 37).

Comment:

In Section 5. Authors claim: the papers are grouped into five main categories ac-cording to their field (cognitive load, immersion, spatial awareness, interaction with the digital environment and attention) and for each category the objectives and outcomes, data preprocessing and artifact removal methods, signal analysis and statistical analysis or classification methods are analyzed. From my point of view, these data in subsections (x.x.3 to x.x.5) are just mentioned and grouped. Since this information exist in the tables, the subsections for the preprocessing and artifact removal methods, signal analysis and statistical analysis or classification methods, have to be removed. Real analysis of data is presented in Figures and in the tables in Section 5 (Tables 12, 13 ,14).

Answer:

We have carefully considered your comment. We believe that these subsections act as a bridge between the tables and the main text and help potential readers, especially those who may not be experts in the field, to gain a more comprehensive understanding of our field of research, facilitating a smoother flow of information.

Comment:

Some of references have DOI, some not.

Answer:

Following the journal template, DOIs were removed from the references (pages 49-56)

Comment:

Minor editing of English language required

Answer:

The manuscript has undergone additional proofreading, and minor mistakes have been corrected.

Round 2

Reviewer 1 Report

The figures are not aligned with the text.

1.     The Prisma figure should be Figure 2.

2.     Section 3.1, Publication Year. The authors write: “Included articles were published from 2013 until December, 2022. The mean publication year of these studies was 2019.9 (median = 2020; SD = 1.8).“ It does not make sense to write “The mean publication year…” I would suggest the following “Included articles were published from 2013 until December, 2022, with incremental number of publications during the years.

3.     In the same section. You either use the Figure or the table. Both are redundant. Besides, the table and figure do not correspond. The Figure number needs updating. The figure includes 2012, which you did not looked at. The numbers of publication in year 2021 and 2022 are exchanged. The Y axis is missing the legend. Decide on the figure or the table and refer to that in the text of the article.

4.     Same for section 3.2, that is, decide on the figure or the table. Make sure that you give them the appropriate figure or table number.

Author Response

We would like to thank the reviewer for the time and effort dedicated on reviewing our paper.

Comment:

The Prisma figure should be Figure 2.

Answer:

The numbering of the figures in the paper has been corrected.

Comment:

Section 3.1, Publication Year. The authors write: “Included articles were published from 2013 until December, 2022. The mean publication year of these studies was 2019.9 (median = 2020; SD = 1.8).“ It does not make sense to write “The mean publication year…” I would suggest the following “Included articles were published from 2013 until December, 2022, with incremental number of publications during the years.“

Answer:

We have removed the sentence and replaced it according to this comment. (page 5, lines 171-172)

Comment:

In the same section. You either use the Figure or the table. Both are redundant. Besides, the table and figure do not correspond. The Figure number needs updating. The figure includes 2012, which you did not looked at. The numbers of publication in year 2021 and 2022 are exchanged. The Y axis is missing the legend. Decide on the figure or the table and refer to that in the text of the article.

Answer:

Based on your suggestion, we have excluded figure 3 (page 5) and the numbering of the remaining figures in the paper has been corrected.

Comment:

Same for section 3.2, that is, decide on the figure or the table. Make sure that you give them the appropriate figure or table number.

Answer:

Based on your suggestion, we have excluded figure 4 (page 5) and the numbering of the remaining figures in the paper has been corrected.

Reviewer 5 Report

The manuscript has been sufficiently improved to warrant publication in BDCC.

Table 3. (Digital environment type and equipment) and Table 4. (EEG equipment) are more illustrative.

The edited in Tables "Main findings" now sound  scientifically.

Conclusion is properly rewritten.

Author Response

Thank you for taking the time to review our paper. Your feedback has greatly improved the quality of our work.